

# Soil microbial biomass and function are altered by 12 years of crop rotation

Marshall D. McDaniel* and A. Stuart Grandy

Department of Natural Resources and the Environment, University of New Hampshire, Durham,

NH USA

*Corresponding Author:

Current address:

Department of Agronomy, Iowa State University

2517 Agronomy Hall

716 Farm House Lane

Ames, IA 50011

Phone: (515) 294-7947

Email:  marsh@iastate.edu



**Abstract**

2        Agriculture-driven declines in plant biodiversity reduce soil microbial biomass, alter

microbial functions, and threaten the provisioning of soil ecosystem services.  We examined
whether increasing temporal plant biodiversity (by rotating crops) can partially reverse these
trends and enhance microbial biomass and function.  We quantified seasonal patterns in soil
microbial biomass, respiration rates, extracellular enzyme activity, and catabolic potential three
times over one growing season in a 12-year crop rotation study at the W.K. Kellogg Biological
Station LTER. Rotation treatments varied from one to five crops in a three-year rotation cycle,
but all soils were sampled under corn to isolate historical rotation effects from current crop
effects.  Inorganic N, the stoichiometry of microbial biomass and dissolved organic C and N
varied seasonally, likely reflecting fluctuations in soil resources during the growing season.
Soils from biodiverse cropping systems increased microbial biomass C by 28-112% and N by
18-58% compared to monoculture corn.  Rotations increased potential C mineralization by as
much as 64%, and potential N mineralization by 62%, and both were related to substantially
higher hydrolase and lower oxidase enzyme activities. The catabolic potential of the microbial
community, assessed with community-level physiological profiling, showed that microbial
communities in monoculture corn preferentially used simple substrates like carboxylic acids,
relative to more diverse cropping systems.  By isolating plant biodiversity from differences in
fertilization and tillage, our study illustrates that crop biodiversity has overarching effects on soil
microbial biomass and function that last throughout the growing season.  In simplified
agricultural systems, relatively small increases in plant biodiversity have a large impact on
microbial community size and function.



**Keywords**: crop rotation; agriculture biodiversity; soil carbon; soil nitrogen; nitrogen mining; community-level physiological profile; mineralization; extracellular enzymes; soil microbial biomass



## Introduction

Research manipulating aboveground biodiversity in grasslands has shown a strong link
between plant species richness and soil functions (Tilman et al. 1997, Zak et al. 2003, Eisenhauer
et al. 2010, Mueller et al. 2013).  While this research has contributed to our understanding of
aboveground-belowground biodiversity in natural ecosystems, it fails to capture the biodiversity
dynamics in agroecosystems, where crop rotations can be used to substitute temporal for spatial
biodiversity.  Given that species richness at any given time in a rotated cropping system is one,
the aboveground-belowground relationships dependent on diversity in agroecosystems and
spatially diverse ecosystems (e.g. grasslands) may not be the same.
Crop rotations have been shown to have large positive effects on soil C, N, and microbial
biomass (McDaniel et al., 2014a), plant pathogen suppression (Krupinsky et al. 2002), and yields
(Ret al. 2008, Riedell et al. 2009). These positive effects of crop production have been
colloquially referred to as the "rotation effect."  However, the mechanistic processes that link
aboveground crop rotational biodiversity and belowground soil processes and contribute to the
"rotation effect" remain elusive.  One hypothesis explaining the benefits of crop rotations is that
greater diversity of plant inputs to soil organic matter (SOM) over time enhances belowground
biodiversity and soil ecosystem functioning (Hooper et al., 2000; Waldrop et al., 2006; Grandy
and Robertson, 2007).  Despite being low in spatial diversity, crop rotations have been shown to
increase soil microbial and faunal biodiversity (Ryszkowski et al. 1998, Wu et al. 2008, Tiemann
et al. 2015) and increase microbial carbon use efficiency (Kallenbach et al. 2015).  One essential
function of soil microbial communities is the catabolism of newly added substrates from crops.
The range and efficiency of microbial catabolism has great implications for ecosystem services



such as sequestering C and soil fertility (Carpenter-Boggs 2000; Kallenbach et al. 2015), but also
for ecosystem "dis-services" such as emission of soil-to-atmosphere greenhouse gases
(McDaniel et al. 2014b).  Furthermore, the partitioning of resources used in catabolism of
residue and formation of SOM will likely affect long-term soil fertility (Lange et al. 2015;
Kallenbach et al. 2015).

Soil microbial catabolic function can be measured using community-level physiological

profiles (CLPP), sometimes referred to as catabolic response profiles.  The basic method for
measuring soil CLPP involves adding a suite of C substrates to soils and measuring the catabolic
response as $CO_2$ production or $O_2$ consumption with redox indicators (i.e. Biolog; Guckert et al.
1996).  These C substrates are typically ecologically-relevant compounds found in soils, and are
intended to represent root exudates, microbial or plant cell structures, or other more-processed
soil organic molecules.  By measuring the respiration response, we can establish a catabolic
"fingerprint" to distinguish soil microbial communities from one another by how they catabolize
these new C substrates.  We can also derive catabolic evenness of that community – a measure of
biodiversity.  Modern agriculture's use of monocultures could have unknown consequences for
soil microbial catabolism and related processes such as SOM mineralization, but to date the
effect of rotation practices and crop diversity on soil microbial functioning remains poorly
understood.

Considering the lack of a clear understanding of how soil microbial functions are

influenced by crop rotations, we sought to examine their effects on soil microbial biomass and
function.  We measured soil microbial catabolic potential, C and N mineralization, extracellular
enzyme activities, and microbial biomass three times over one growing season in a long-term



crop rotation experiment at the W.K. Kellogg Biological Station (est. 2000). All soils were
measured under the same crop, allowing us to separate historical rotation from current crop
effects. We hypothesized that soils under more diverse crop rotations would show greater
catabolic diversity and have higher measures of soil function (enzyme activities, soil microbial
biomass, potentially mineralizable C and N). In addition, we hypothesized that crop rotation
effects would vary seasonally, being greatest in the spring and lessen over the growing season
with the emerging influence of the current crop. The rationale for this second hypothesis is that
early in the season all soils are coming out of different crops from the previous year, but over the
growing season under corn the soils will become more functionally similar as the immediate crop
has greater influence. Alternatively, significant Rotation by Season interactions on soil
microbial functioning that do not converge over the growing season point to historical effects of
rotations on differences in soil microbial communities and SOM.
**Materials and Methods**

This study was conducted in the Cropping Biodiversity Gradient Experiment (CBGE) at

the W.K. Kellogg Biological Station Long-term Ecological Research site (42° 24', 85° 24' W).
The CBGE was established in 2000 and consists of crop rotations ranging from monocultures to
a 5-species rotation (http://lter.kbs.msu.edu/research/long-term-experiments/biodiversity-
gradient/). The crop rotations were repeated but with different rotation phases within all four
blocks. For example, the corn-soy-wheat rotation is replicated three times within each block, but
these replicates are planted to a different crop each year. The plot dimensions were 9.1 x 27.4 m
and received the same chisel plow tillage to a depth of approximately 15 cm, and received no
inputs (e.g. pesticides or fertilizers) that would have confounded the treatment effects of
rotational diversity (Smith et al. 2008). Mean annual temperature and precipitation at the site are




9.7°C and 890 mm, respectively.  The two main soil series located at the site are Kalamazoo, a
fine-loamy, mixed, mesic Typic Hapludalf, and Oshtemo, a coarse-loamy, mixed, mesic Typic
Hapludalf  (KBS, 2012).  Soil pH in the top 10 cm ranges from 4.9 to 6.1 (1:1 v of 0.01 M
$CaCl_2$).

Soils were collected from the following cropping systems: monoculture corn (*Zea maize*

*L.,* mC), corn-soy (*Glycine max,* CS), corn-soy-wheat (*Triticum aestivum,* CSW), corn-soy-
wheat with red clover cover crop (*Trifolium pretense,* CSW1), and corn-soy-wheat with red
clover + rye cover crops (*Secale cereale,* CSW2).  Soil sampling took place on April 27[th], 2012;
July 19[th], 2012; and November 1[st], 2012 – hereafter referred to as spring, summer, and autumn.
Corn was planted in all plots on June 11[th] 2012.  Three 5 cm soil cores (0-10 cm deep) were
collected within each plot, homogenized in the field, and then put on ice and shipped to the
University of New Hampshire.  In the lab, field-moist soils were immediately sieved using a 2
mm sieve.  A sub-sample was taken from sieved soil and dried at 105 °C to determine
gravimetric water content.  Water-holding capacity was determined as the water content after
soils were saturated and drained for 6 h.

*Soil carbon and nitrogen parameters*

Five g of field-moist soil were extracted for inorganic N with 40 ml of 0.5 M $K_2SO_4$.  The

soil slurries were shaken for 1 h before the extracts were filtered on Whatman GF/C (5) filters
and filtrate frozen and stored until analysis.  Soil nitrate ($NO_3^-$) and ammonium ($NH_4^+$) were
measured using the methods detailed in McDaniel et al. (2014c).  We also used the same extracts
to measure dissolved organic C and N (DOC and DON).  The extracts were run on a TOC-TN
analyzer (TOC-V-CPN; Shimadzu Scientific Instruments Inc., Columbia, MD, USA).  Total C





and N were analyzed by sieving soils through 2 mm sieve, grinding and analyzing on an ECS
4010 CHNSO Elemental Analyzer (Costech Analytical Technologies, Inc., Valencia, CA).

Potential mineralization rates of C (PMC) and net N (or PMN) estimate the quantity of

potentially-mineralizable SOM at an optimal temperature and soil moisture, and reflect both the
activity of the microbial community and availability of SOM (Paul et al. 1999; Robertson et al.
2000).  These mineralization assays provide a good indicator of the potential for a soil to provide
plants with N (Stanford and Smith 1972, Robertson et al. 1999).  Both PMC and PMN were
measured on air-dried soils that were placed into Wheaton serum vials and brought to 50%
water-holding capacity and incubated for 6 months.  During this 6-month period $CO_2$ efflux was
measured on a LI-820 infrared gas analyzer (LI-COR, Lincoln, NE).  Efflux was measured using
the change in headspace $CO_2$ concentration measured between two time points.  Each soil efflux
measurement began by aerating jars, capping, and injecting a time-zero sample and then a second
sample between 5 hours up to 2 days later.  Efflux was calculated as the difference in $CO_2$
concentration between the two time points divided by time.  Measurements of PMC occurred
more frequently at the beginning of the experiment (daily) and became less frequent toward the
end (once every other week), for a total of 19 sampling events over 120 days.  The PMN was
assessed by extracting the inorganic N produced at the end of the incubation with the methods
described above.
*Soil microbial parameters*

Soil microbial biomass C and N were determined using the modified chloroform

fumigation and extraction method (Vance et al. 1987), but modified for extraction in individual
test tubes (McDaniel et al. 2014c).  Briefly, two sets of fresh, sieved soil (5 g) were placed in 50
ml test tubes, and 1 ml of chloroform was added to one set of tubes and capped.  The tubes sat



overnight (24 h) and were then uncapped and exposed to open air in a fume hood to allow
chloroform to evaporate. Soils were then extracted in the tubes with 25 ml of 0.5 M $K_2SO_4$. The
chloroform fumigated and non-fumigated extracts were run on a TOC-TN analyzer (TOC-V-
CPN; Shimadzu Scientific Instruments Inc., Columbia, MD, USA). We used 0.45 (Joergensen
1996) and 0.54 (Brookes et al. 1985) for the C and N extraction efficiencies.
Soils were analyzed for 7 extracellular enzyme activities (EEAs): β-1,4-glucosidase
(BG), β-D-1,4-cellobiohydrolase (CBH), β-1,4-N-acetyl glucosaminidase (NAG), acid
phosphatase (PHOS), Tyrosine aminopeptidase (TAP), Leucine aminopeptidase (LAP),
polyphenol oxidase (PO), and peroxidase (PER). Given the large number of samples (60) and
variety of measurements made at each of 3 sampling dates, soil EEAs were conducted on frozen
samples within 4 weeks of sampling. Extracellular enzyme activity assays were carried out
following previously published protocols (Saiya-Cork et al. 2002, German et al. 2011), but with
some modifications. Briefly, 1 g of soil was homogenized with a blender in 80 ml of sodium
acetate buffer at pH 5.6 (the average pH at the site). Soil slurries were pipetted into 96-well
plates and then analyzed on a Synergy 2 plate reader (BioTek Instruments, Inc., Winooski, VT).
For oxidoreductase enzymes, the supernatant from the slurry plates were pipetted into a clean
plate to avoid interference with soil particles. Hydrolase assays were read at 360/40 and 460/40
fluorescence and oxidoreductases at 450 nm absorbance. For more details on the extracellular
enzyme methods see McDaniel (2014c).
Community-level physiological profiles (CLPP) were conducted using the MicroResp[TM]
system (Chapman et al. 2007, Zhou et al. 2012, McDaniel et al. 2014b). The MicroResp[TM]
system allows for high-throughput measurement of soil catabolic responses to multiple C
substrates. Each soil was loaded into 96 deep-well plates using the MicroResp[TM] soil dispenser,



and then brought to 50% water-holding capacity. Thirty-one substrates were used at
concentrations ranging from 7.5 to 30 mg C per g of soil $H_2O$, as recommended by the
MicroResp$^{TM}$ manual (Table S1). Soil and substrates were combined in analytical triplicates and
a $CO_2$ detection plate (agar containing creosol red) was immediately placed onto the deep-well
plate with an air tight seal provided by the MicroResp$^{TM}$ kit. The soil and substrates were
incubated in the dark for 6 h at 25 °C. The detector plate absorbencies were read at times 0 and
6 h at 540 nm on a Synergy 2 plate reader (BioTek Instruments, Inc., Winooski, VT).
Absorbance data were normalized and converted to a $CO_2$ efflux rate ($\mu$g $CO_2$-C g soil$^{-1}$ h$^{-1}$),
according to the MicroResp$^{TM}$ procedure (Chapman et al. 2007).
*Data analyses*
Cumulative potentially mineralizable C and N were calculated in SigmaPlot v12.5 (Systat
Software, Inc., San Jose, CA) using the integration macro, *area below curves.* Data not
conforming to ANOVA assumptions of homogeneity of variances and normality were
transformed before analyses (Zuur et al. 2010). Catabolic evenness (CE), a measure of substrate
diversity, was calculated using the Simpson-Yule index, $CE = 1/\Sigma p_i^2$, where $p_i$ is the proportion
of a substrate respiration response to the total response induced from all substrates (Degens et al.
2000, Magurran 2004). Metabolic quotient was calculated simply as the basal respiration over 6
h (determined in the MicroResp$^{TM}$ method) divided by the MBC.
Response variables were analyzed using a 2-way analysis of variance (ANOVA), with
Season and Rotation as main effects. The ANOVAs were conducted in SAS 9.3 (SAS Institute,
Cary, NC) using the *proc mixed* function and post-hoc *t* tests were used to determine significant
differences among means using *ls means*. Block was assigned as a random effect variable within





the model.  Correlations between variables were made using *proc corr*, and Pearson's correlation
coefficients are reported.  Model effects were deemed significant if $\alpha < 0.05$.

All multi-variate data analyses were performed with R software (The R Foundation for

Statistical Computing, Vienna, Austria).  CLPP data were checked to ensure they conformed to
principal components analysis assumptions.  The *prcomp* function in the *vegan* package
(Oksanen et al. 2016) was used for PCA of CLPP data.  In order to correlate environmental
variables with the multi-variate CLPP data we used the *envirfit* function.
**Results**

It was a relatively dry year at the KBS-LTER in 2012, which had an annual precipitation

of 742 mm, compared to the historical mean of 870 mm (Hamilton et al. 2015).  There was also
an anomalous warm spell in mid- to late-March (Fig. S1).  After harvest, the corn yield (kg ha$^{-1}$ $\pm$
SE) in each treatment was as follows:  mC = 2846 $\pm$ 152, CS = 4208 $\pm$ 575, CSW = 4107$\pm$ 220,
CSW1 = 4015 $\pm$ 187, CSW2 = 5219 $\pm$ 1180 (KBS-LTER 2015).
*Soil C and N biogeochemistry*

There were no significant Rotation or Season effects on total soil C and N, although both

soil C and N tended to increase with the number of crops in rotation (Table1).  Soil NO$_3^-$-N was
the only variable that showed a significant Season X Rotation interaction ($P < 0.001$).  Seasonal
soil NO$_3^-$-N concentrations were highest in summer (10.33) followed by spring (2.98), and
autumn (1.28 mg kg$^{-1}$).  Soil NH$_4^+$-N was generally low, but summer had more than twice the
concentrations of spring and autumn.  Dissolved organic C (DOC) and N (DON) was very
dynamic over the year.  The DOC was highest in the autumn, while DON was over six times
greater in the summer than the other seasons ($P$s $< 0.001$).  The mean DOC:DON in autumn was



22.5, twice that of spring and 13 times that of the summer.  There were significant crop rotation
effects on $NO_3^-$-N, DOC, and DON.  During the summer the two cover crop treatments had $NO_3^-$
concentrations 67% greater than CSW and CS treatments, and 158% greater than mC.  The
CSW1 treatment had 112% greater DOC concentrations than mC ($P < 0.001$), and two cover
crop treatments had 107% greater DON than non-cover crop treatments and 211% more than the
mC treatment. The potentially mineralizable pools of C and N showed significant main effects of
both Season and Rotation ($P < 0.03$), but no interactions.  Generally, both PMC and PMN
increased with increasing number of crops in rotation (Fig. 1), and PMC was highest during the
autumn, while PMN was highest during the summer.
*Soil microbial dynamics*

The mean soil microbial biomass C (MBC) was 332 µg C g soil$^{-1}$ across all seasons and

crop rotations, but both Season ($P < 0.001$) and Rotation ($P = 0.008$) had significant effects on
MBC (Fig. 2).  Soils collected in autumn had more than twice the MBC than those collected in
spring and summer.  Microbial biomass C was increased by increasing crop diversity across all
seasons (Fig. 2).  Increasing the number of crops in rotation increased MBC on average by 28,
113, 112, and 72% in the CS, CSW, CSW1, CSW2 rotations, respectively, compared to mC
(across all seasons).  Microbial biomass N (MBN) also showed both Season ($P < 0.001$) and
Rotation ($P = 0.005$) effects, but no interaction.  These effects were strongest in the spring and
summer (Fig. 2), but also showed an increase with increasing number of crops.  Increasing the
number of crops in rotation increased MBN on average by 18, 58, 54, and 50% in the CS, CSW,
CSW1, and CSW2 compared to mC (across all seasons).  Microbial biomass C:N showed a
significant interaction ($P = 0.013$), with more diverse cropping systems having greater
MBC:MBN in summer and autumn, but not spring.  The metabolic quotient ($qCO_2$), is often used



as a proxy for microbial respiration efficiency (Anderson & Domsch 1989, 2010; Wardle &
Ghani 1995).  Season ($P < 0.001$) and Rotation ($P = 0.006$) both influenced $qCO_2$, with increased
crop diversity decreasing the $qCO_2$ by 16, 40, and 28 % in CSW, CSW1, and CSW2 compared to
mC.  However, the CS rotation increased $qCO_2$ by +15 % $qCO_2$ compared to mC (Fig. 2).

Soil extracellular enzymes were very dynamic over the three seasons, as evidenced by

radar plots in which the area and shape for each treatment change quite drastically over the
growing season (Fig. 3).  A MANOVA with all eight EEAs showed significant Season ($P <$
0.001) and Rotation ($P < 0.001$) main effects, but no interaction.  Most individual enzymes
showed only significant Rotation effects except for PO, which also showed a significant Season
effect with autumn greater than the other seasons (Table 2).  The soil enzyme responsible for
cleaving a glucosamine from chitin (NAG) and the lignin-reducing enzyme that uses peroxide
(PER) were the only enzymes that showed a significant Season X Rotation interaction ($P <$
0.001).  Spring had the greatest activities of LAP, 175% greater than the average of the other
seasons (Fig. 3, Table 2).  In summer, we see a shift to the highest PHOS activity – 25% greater
than spring and 99% greater than autumn.  Season had no effect on BG or CBH but showed
significant main effects of rotation, with the CSW1 treatment having an average of 42 and 50 %
higher BG and CBH activity than CS and mC soils, respectively.  The majority of the hydrolase
enzymes were higher in the cover crop treatments compared to that of the non-cover crop
treatments, especially mC (Table 2, Fig. 3).  The two oxidoreductase enzymes (PO and PER)
decreased with crop diversity.  There were no significant main effects on the enzyme ratio used
to assess C-versus-N demand (BG to NAG+LAP).

The community-level physiological profile (CLPP), a catabolic profile of the soil

microbial communities, showed both significant Season ($P < 0.001$) and Rotation ($P = 0.003$)




main effects (Figs. 4, S2; Table 3).  A principal components analysis of the CLPP data showed
that the summer soils corresponded with highest carboxylic acid utilization (Fig. 4), as Season
was the strongest discriminating factor along principal component 1 (PC1, Table 3).  However,
when rotating and examining PC2 and PC3, there was a strong treatment gradient from the
bottom right to upper-left quadrants of the graph (Fig. 4, right panel).  The lower-diversity
treatments corresponded with greater use of carboxylic acid substrates.  Across seasons, summer
exhibited the lowest catabolic evenness (12.9), but there was no crop rotation effect on catabolic
evenness using all substrates (Table 4).

Due to the overwhelming influence of carboxylic acids in the PCA variation, and their

possible role in abiotic reactions leading to $CO_2$ emissions (Maire et al. 2012, Pietravalle and
Aspray 2013), we split the 31 substrates into two sets to analyze separately: 1) Non-carboxylic
acid substrates – a total of 21 substrates, and 2) carboxylic acids by themselves – 10 substrates.
Season, again, was a dominant significant effect on the MANOVAs in both groups of substrates
($P$ values < 0.001, Fig. S3).  The non-carboxylic acid CLPP showed a significant treatment effect
with PC1 and PC2, and clear separation between low and high diversity cropping systems ($P =$
0.012, Fig. S3).  The monoculture corn, and lower diversity treatments, associated with more
complex substrates.  In the carboxylic acid CLPP there was also a significant treatment effect,
but with PC2 and PC3, and clear separation between low and high diversity cropping systems
along PC3 ($P = 0.035$, Fig. S3).  Interestingly, the lower diversity (especially monoculture corn),
were more associated with simple carboxylic acids (Cit, Mlo, and Mli) on the positive half of
PC3.  When carboxylic acids were split from the substrates, crop rotation had a significant effect
on catabolic evenness – decreasing the catabolic evenness both within non-carboxylic acids and
carboxylic acids by as much as 4 and 13% respectively (Table 4).





We used the soil microbial responses of EEA and the CLPP because we assumed they
would be complimentary.  Indeed, this was the case.  Measuring NAG enzyme and adding the
Nag amine to the soils showed a somewhat tight relationship, but this relationship was not
constant over the seasons.  More specifically, the NAG enzyme was quite higher in the autumn
compared to summer and spring, and showed a steeper linear relationship with the $CO_2$ response
after adding the Nag amine to soils (Fig. S4).  Additionally, when the CLPP substrates were
grouped by guild and correlated with EEA there were strong relationships (Fig. S5).  For
example, total amino acid catabolic response positively correlated well with LAP+TAP enzymes
($r^2 = 0.35$, $P < 0.001$) meaning that high activity of these enzymes in soils corresponded with
high relative use of these substrates when added to soils, compared to other substrates added to
the soil.  This suggests that the LAP and TAP enzymes strongly reflect demand for N-bearing
amino acids in soils.  However, the catabolic response of the 'Complex' guild was negatively
correlated with PO ($r^2 = 0.29$, $P < 0.001$).
*Relationships between soil biogeochemical factors, microbial functioning and yield*
Over the three seasons many soil biogeochemical factors correlated with microbial
catabolic potential, both with individual C substrate guilds and catabolic evenness (Table 5).
Abiotic factors such as pH and sand content correlated with the specific use of particular
substrates. Soil pH positively correlated with N-containing and complex substrates, but strongly
negative with carboxylic acids.  Sand content negatively correlated with amino acids and
carbohydrates, but positively with carboxylic acids.  The microbial response to amino acids and
amines correlated best with $NO_3^-$-N and many of the specific enzyme activities, showing
negative relationships which indicated a linkage between demand for N and usage of N-bearing
substrates (i.e. when supply is high, demand and usage of N substrates is low).  Soil $NO_3^-$-N was



also significantly negatively correlated with catabolic evenness.  We used the soil microbial
responses of EEA and the CLPP because we assumed they would be complimentary.  For
example, adding N-acetyl glucosamine in the CLPP should be related to ß-1,4-N-acetyl
glucosamindase (NAG) enzyme activity.   Indeed, this was the case.  Measuring NAG enzyme
and adding the Nag amine to the soils showed a somewhat tight relationship (Fig. S4).
Additionally, when the CLPP substrates were grouped by guild and correlated with EEA there
were strong relationships (Fig. S5).  For example, total amino acid catabolic response positively
correlated well with LAP+TAP enzymes ($r^2$ = 0.35, $P$ < 0.001) meaning that high activity of
these enzymes in soils corresponded with high relative use of these substrates when added to
soils, compared to other substrates added to the soil.  This suggests that the LAP and TAP
enzymes strongly reflect demand for N-bearing amino acids in soils.  However, the catabolic
response of the 'Complex' guild was negatively correlated with PO ($r^2$ = 0.29, $P$ < 0.001).  Soil
PMN was better correlated with crop yields ($r^2$ = 0.61) than $NO_3^-$ in early spring (Fig. S6).
**Discussion**

Increasing biodiversity in this long-term crop rotation experiment has altered the soil

microbial dynamics across an entire growing season.  This is despite the fact that the soils in our
study were all under the same crop (corn) for the season, indicating that observed differences
among soils reflect long-term rotation effects rather than the current crop.  Microbial biomass C,
N, potential mineralization, and catabolic potential were all altered by crop rotations, although
the rotation effect for some of these indicators of microbial functioning also depends upon the
season.  Soil microbial biomass and activity are now widely recognized as pillars of soil health
(Doran and Zeiss 2000).  Our results clearly indicate that practices like diversifying
agroecosystems (through crop rotations) enhances this aspect of soil health, and this is also likely



linked to changes in SOM dynamics (Tiemann et al. 2015) as well as the observed differences in
yield among crop rotations (Fig. S6).
*Seasonal dynamics and N limitation*
Season strongly influenced the measured pools of labile C and N (Table 1), as well as the
microbial biomass size and functioning within this agroecosystem (Figs. 1-4). We showed that
the greatest microbial biomass and activity occurred in autumn, but that potentially mineralizable
N peaked in summer. In perennial and annual cropping systems in Iowa, potentially
mineralizable N declined from spring to late summer (Hargreaves and Hofmockel 2013); in
addition, extracellular enzyme activities peaked in July but there was little effect of the cropping
system. Season was shown to affect microbial biomass and potentially mineralizable C and N
pools in a wheat-sorghum-soybean rotation in south-central Texas (Franzluebbers et al. 1994,
1995, Franzluebbers 2002), but timing for peak values differed depending on the study and
cropping systems, likely reflecting different climates and soil types. The frequently observed
late-summer spike in microbial biomass and activity may be related to higher temperatures
during this time period; however, even within agroecosystems, the timing for maximal microbial
biomass varies substantially, although few microbial maxima are reported in winter (Wardle,
1992). Our findings highlight the dynamic nature of soil microbial biomass and activity,
especially with regards to the supply and demand of N (e.g. microbial C:N, substrate utilization,
and extracellular enzyme activities), which is likely a limiting nutrient in these agroecosystems
that are receiving no exogenous N inputs.
The summer warrants discussion because the sample was collected after a prolonged
period of hot and dry days, but right after a large rainfall event. This rainfall event (> 18 mm d$^{-1}$,





Fig. S1) increased the volumetric water content in the 0-10 cm of a nearby soil by over 54%
from the lowest value of the year (0.1, data shared from Hamilton et al. 2015), and we know
from previous research that drying-wetting cycles are important soil biogeochemical drivers
(Borken and Matzner, 2009) and can alter microbial structure and functioning (Fierer et al. 2003,
Schimel et al. 2007, Tiemann and Billings 2011, McDaniel et al. 2014b).  Indeed, the summer
showed several signs of the soil microbial community being impacted by a rapid dry-wet event:
lower overall microbial biomass C, extremely high $NO_3^-$-N concentrations (Table 2), high
potential N mineralization (Fig. 1), high extracellular enzyme activities per unit of microbial
biomass (Fig. S2, presumably a result of lysed intracellular enzymes, Burns et al. 2013), and the
particularly strong response of the summer soils to carboxylic acids (a highly-labile class of
compounds used by fast-growing, opportunistic microbes, that would be found after a
disturbance such as a dry-wet event, Figs. 4 and S3).  Dry-wet cycles may drive microbial C and
N to be reallocated to stress-response compounds instead of growth or reproduction, making C
and N more vulnerable to loss from soils (Schimel et al. 2007).  We captured one of these dry-
wet events during one of the driest summers in the Kellogg Biological Station LTER's history
and we show high soil inorganic N concentrations and altered microbial dynamics relative to the
other dates.  Climate change may increase the frequency and magnitude of these rapid dry-wet
cycles (Groffman et al. 2001, McDaniel et al. 2014d), and thus may have long-term impacts on
soil microbial functioning and biogeochemistry.

In the autumn we found several lines of evidence that indicate soil microbes are N, rather

than C, limited.  These lines of evidence include: lowest soil inorganic N concentrations, low
potentially mineralizable N, high microbial biomass C:N and DOC:DON ratios, and high TAP
and NAG enzymes relative to other enzymes (although interestingly not LAP), and finally strong



respiration response to the addition of amines and amino acids (Fig. 4).  The unusually high
microbial biomass C:N in autumn was very surprising (mean of 18.2 versus 7.5 and 9.9 in spring
and summer), but microbial biomass C:N has been known to reach as high as 30 in laboratory
conditions (Schimel et al. 1989).  Additionally, the few days before and after the collection of the
autumn sample were unusually cold (Fig. S1), and cold temperatures and freezing can cause
accumulation of carbohydrates in fungi (Tibbits et al. 2002), which could widen microbial C:N
ratio.  Regardless of environmental conditions widening microbial biomass C:N, it is likely that
N limitation is a major factor considering this experiment receives no fertilizer N for 12 years.
*Crop biodiversity and soil microbial functioning*

Both soil microbial biomass and functioning were strongly affected by increased crop

diversity through rotation.  This rotation effect on soil microbial biomass and functioning were
largely independent of the season, as indicated by the limited number of observed Season X Crop
Rotation interactions.  The exception to this was microbial biomass C:N and two extracellular
enzyme activities (NAG and PER), which are likely indicative of the enhanced ability of soil
microbes under diverse rotations to process, provision, and retain N.  These findings are
consistent with plant biodiversity studies which find increased aboveground diversity enhances
soil microbial biomass and functioning in natural (Stephan et al. 2000, Zak et al. 2003, Lange
2015) and agricultural ecosystems (Lupwayi et al. 1998; Xuan et al. 2012; Tiemann et al. 2015).

In our study, the strong response of soil microbial functioning to crop diversity is

particularly interesting given that all treatments were under the same crop (corn) over the entire
growing season.  Some studies have shown that the current plant species identity often trumps
biodiversity legacy in controlling belowground microbial structure and functioning (Stephan et



al. 2000, Wardle et al. 2003, Bartelt-Ryser et al. 2005). Conversely, several studies have pointed
to weak or no influence of current plant species on soil microbial structure and functioning
(Costa et al. 2006, Kielak et al. 2008). The question of whether plant species identity versus
spatial and temporal diversity has a stronger control on soil biota remains a critical question in
terrestrial ecology. Here we show strong evidence for a biodiversity "carry-over" effect (Bartelt-
Ryser et al. 2005), where the legacy of crop rotation affects soil microbial biomass and
functioning of the current year, even though the soils are all planted under the same crop species.
We hypothesized that increasing biodiversity in agroecosystems through rotation would
result in soil microbial communities that are more diverse, and thus would more evenly use
added C substrates (i.e. increase catabolic evenness, or decrease the variation in use among
substrates). This hypothesis stems from the "plant diversity hypothesis", which posits that soil
community and functional biodiversity is linked to plant biodiversity mostly through the
diversity of plant inputs to SOM (Lodge, 1997; Hooper et al., 2000; Waldrop et al., 2006;
Korboulewsky et al., 2016). However, in our study, we found no evidence that crop rotational
diversity increased overall soil catabolic evenness. There is some evidence that crop rotations
can alter soil bacterial catabolic diversity, or the ability to use different C substrates (Lupwayi et
al. 1998, Larkin 2003, Govaerts et al. 2007), however all of these studies used Biolog which has
several limitations (Preston-Mafham et al. 2002). The MicroResp™ system's main benefit is
that it adds C substrates directly to the soil instead of tranferring an inocullum from a soil slurry.
The discrepancy between our study and these studies other studies may be due to methodological
differences between Biolog and MicroResp™. Our lack of evidence for an aboveground-
belowground link to catabolic potential aligns with findings from other studies that have found



functional diversity measures of soil microbes are not related to plant diversity (Bartelt-Ryser et
al. 2005, Jiang et al. 2012), nor plant species in general (McIntosh et al. 2013).
However, it is important to note that in our study when a subset of the C substrates were
analyzed (all non-carboxylic), we found that increased crop diversity actually decreased
catabolic evenness.  This is unusual considering soils from this same study, but collected a year
prior, showed increases of soil biodiversity (Shannon-Weiner index or H') with increased crop
diversity when measuring phospholipid fatty acids (Tiemann et al. 2015); and diversity has been
found to be strongly, positively related to evenness in plants and animals (Stirling & Wilsey
2001).  Our finding of no change in (or lower) catabolic evenness with increasing crop
biodiversity is also contradictory to the findings of Degens et al. (2000), whom showed that
management practices that decrease soil C are associated with low catabolic evenness since we
show a general trend of increasing soil C.
Perhaps the incongruity between the positive effect of crop rotation on H', but slightly
negative effect on catabolic evenness, could lie in the difference between who is targeted by the
catabolic response profile.  Most bacteria are thought to be generalists with regard to the use of C
substrates (Mou et al. 2008; Goldfarb et al. 2011), whereas fungi may show more specialization
(Hanson et al. 2008; Treseder et al. 2015).  Not to mention the catabolic response method used
here is probably more favorable to detecting response of bacteria rather than fungi, because fungi
are more sensitive to the disturbance of preparing the soils (Frey et al., 1999) and bacteria have
faster growth and reproduction (i.e. respiration was measured for only 6 h).  Thus a high
catabolic evenness may not be a good indicator of soil biodiversity in soils with high relative
bacterial biomass and activity, which is typical of many agroecosystem soils (Strickland &
Rousk 2010).  Instead, a low catabolic evenness may actually be better suited to detect C-use




specialization in bacterial-dominated microbial communities.  In support of this idea, we have
evidence from these same soil samples that crop diversity significantly decreased H' for bacterial
16S rRNA by as much as 5 % compared to monoculture corn (Peralta et al. *in preparation*).
These inconsistencies, especially between methods of measuring soil microbial diversity, are
highlighted in a recent quantitative review (Venter et al. 2016) but overall crop rotations tend to
increase soil biodiversity by 3 % and richness by 15 %.  Regardless of belowground diversity
trends, crop rotations did create functionally distinct microbial communities in our study (Fig. 4).
We still do not have a complete understanding of how crop rotations alter soil microbial
diversity, nor (arguably more importantly) how these changes in belowground diversity might
provide beneficial soil ecosystem services like increasing soil C or mineralizing more N to
increase crop yields.

One trend that emerges across the suite of 31 C substrates is that crop rotations altered the

preference for substrates of differing quality (i.e. complex versus simple C substrates).  The soils
from monoculture corn corresponded to greater use of simple C substrates (especially carboxylic
acids), and showed less response to the suite of N-containing and complex substrates.  This
finding corroborates a previous study we conducted using whole-plant residues, in which we
showed diverse crop rotations resulted in greater decomposition of low quality crop residues
(e.g. corn and wheat, McDaniel et al. 2014c).  Further, when looking only within the relatively
labile carboxylic acid substrates, microbial communities in the less diverse crop rotations (mC,
and CS to a lesser extent) responded to more labile, low-molecular weight carboxylic acids (e.g.
citric, malonic, and malic acid), while soil microbes from more biodiverse crop rotations
responded more to complex, higher-molecular weight carboxylic acids (e.g. caffeic, tartaric, and
vanillic acids - Fig. S4). The strong effects of crop biodiversity on both overall usage of and the



types of catabolized carboxylic acids is not surprising due to the small, yet dynamic pool of these
compounds in soil (Strobel 2001). Since soil microbial function (as measured by CLPPs) is an
aggregate measure of both the community composition and available resources, it is impossible
to tease out which (or both) have changed due to increased crop biodiversity. However, our
overall findings indicate that increased aboveground biodiversity through crop rotations and
cover crops appears to facilitate soil microbial communities' use of complex C substrates relative
to simple ones.
*Conclusions*

As the growing population is increasingly reliant on soils for food, fiber, and fuel we will

either need to consume less, put more land into production, or better use the land we already
have in production. Putting more land in production will likely result in declines in local and
global biodiversity. Thus, it is critical to incorporate biodiversity through any means possible
into the existing managed ecosystems – even including biodiversity through time as with crop
rotations. Here we show that both microbial biomass and function are strongly influenced by
cropping diversity. In fact, the influence of crop rotations on soil microbes and functioning lasts
over an entire growing season and even when all soils are under the same crop. Crop rotations
clearly enhance soil microbial biomass and activity, which are now considered a pillar of soil
health. Furthermore, this rotation effect on soils also appears to facilitate microbes in supplying
more N to crops (Fig. S6). Overall, our study highlights the importance of incorporating
biodiversity into agroecosystems by including more crops in rotation, especially cover crops, to
enhance beneficial soil processes controlled by soil microbes.
**Acknowledgements**



Support for this research was also provided by the NSF Long-Term Ecological Research
Program (DEB 1027253) at the Kellogg Biological Station and by Michigan State University
AgBioResearch.  We would like to acknowledge both Kay Gross and Phil Robertson who
originally established these sites and have kindly provided our research team with access to
them.  Thanks to Stephen Hamilton and co-authors whom provided soil microclimate data from a
nearby experiment.  Also, we would like to thank Serita Frey for helpful advice dealing with the
CLPP data, and Christopher Fernandez for giving feedback on an early draft of this manuscript.
We are grateful for financial support from the United States Department of Agriculture Soil
Processes Program, grant #2009-65107-05961.

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

Methods controls, and ecosystem implications. Soil Biol. & Biochem. 42:1385-1395.
Thoms, C., Gattinger, A., Jacob, M., Thomas, F. M., and Gleixner, G. 2010. Direct and indirect
effects of tree diversity drive soil microbial diversity in temperate deciduous forest. Soil
Biol. & Biochem. 42:1558–1565.
Tiemann, L. K., and S. A. Billings. 2011. Changes in variability of soil moisture alter microbial
community C and N resource use. Soil Biol. & Biochem. 43:1837–1847.
Tiemann, L. K., Grandy, A. S., Atkinson, E. E., Marin-Spiotta, E., and McDaniel, M. D. 2015.
Crop rotational diversity enhances belowground communities and functions in an
agroecosystem. Ecol. Lett. 18:761-771.
Tilman, D., Knops, J., Wedin, D., Reich, P., Ritchie, M., and Siemann, E. 1997. The influence of
functional diversity and composition on ecosystem processes . Science 277 :1300–1302.
Treseder, K. K., Marusenko, Y., Romero-Olivares, A.L., and Maltz, M.R. 2016. Experimental
warming alters potential function of the fungal community in boreal forest. Global Change
Biol. - doi:10.1111/gcb.13238





Trivedi, P., Rochester, I. J., Trivedi, C., Van Nostrand, J. D., Zhou, J., Karunaratne, S.,
Anderson, I. C., and Singh, B. K.  2015. Soil aggregate size mediates the impacts of
cropping regimes on soil carbon and microbial communities. Soil Biol. & Biochem. 91:169-
673     181

Vance, E. D., Brookes, P. C., and Jenkinson, D. S. 1987. An extraction method for measuring
soil microbial biomass C. Soil Biol. & Biochem. 19:703–707.
Venter, Z.S., Jacobs, K., Hawkins, H.-J. 2016. The impact of crop rotation on soil microbial
diversity: A meta-analysis. Pedobiologia xxx-xxx.
Waldrop, M. P., Zak, D. R., Blackwood, C. B., Curtis, C. D., and Tilman, D. 2006. Resource
availability controls fungal diversity across a plant diversity gradient. Ecol. Lett. 9:1127–
680     1135.

Wardle, D.A. 1992. A comparative assessment of factors which influence microbial biomass
carbon and nitrogen levels in soil. Biol. Rev. 67:321–358.
Wardle, D.A. and Ghani, A. 1995. A critique of the microbial metabolic quotient ($q$CO$_2$) as a
bioindicator of disturbance and ecosystem development.  Soil Biol. & Biochem. 12: 1601-
685     1610.

Wardle, D. A., Yeates, G. W., Williamson, W., and Bonner, K. I. 2003. The response of a three
trophic level soil food web to the identity and diversity of plant species and functional
groups. Oikos 102:45–56.
Wu, T., Chellemi, D., Graham, J., Martin, K., and Rosskopf, E. 2008. Comparison of soil
bacterial communities under diverse agricultural land management and crop production
practices. Microb. Ecol. 55:293–310.
Xuan, D., Guong, V., Rosling, A., Alström, S., Chai, B., and Högberg, N. 2012. Different crop
rotation systems as drivers of change in soil bacterial community structure and yield of rice,
*Oryza sativa*. Biol. Fert. Soils 48:217–225.
Zak, D. R., Holmes, W. E., White, D. C., Peacock, A. D., and Tilman, D. 2003. Plant diversity,
soil microbial communities, and ecosystem function: Are there any links? Ecology
84:2042–2050.
Zhou, X., Wu, H., Koetz, E., Xu, Z., and Chen, C. 2012. Soil labile carbon and nitrogen pools
and microbial metabolic diversity under winter crops in an arid environment. Appl. Soil
Ecol. 53:49–55.
Zuur, A. F., Ieno, E. N., and Elphick, C. S. 2010. A protocol for data exploration to avoid
common statistical problems. Method Ecol. Evol. 1:3–14.



Table 1. Soil carbon (C) and nitrogen (N) pools by season and crop rotation

| Season | Crop Rotation | Total Organic C g kg⁻¹ | Total N | NO₃⁻-N mg kg⁻¹ | NH₄⁺-N | DOC | DON | C:N | DOC:DON |
|---|---|---|---|---|---|---|---|---|---|
| Spring | | | | | | | | | |
| | mC | 8.1 (0.8) | 0.8 (0.1)ab | 2.66 (0.79) | 0.06 (0.01)B | 14 (4)bB | 5 (1)bB | 9.8 (0.3) | 2.8 (0.2)B |
| | CS | 7.8 (1.2) | 0.8 (0.1)ab | 2.97 (1.13) | 0.06 (0.01)B | 11 (1)abB | 5 (1)bB | 10.3 (0.4) | 2.1 (0.2)B |
| | CSW | 7.0 (0.6) | 0.7 (0.1)b | 2.67 (0.39) | 0.10 (0.02)B | 21 (8)abB | 6 (1)abB | 10.4 (0.4) | 4.2 (1.9)B |
| | CSW1 | 8.7 (0.4) | 0.9 (0.1)a | 3.10 (0.66) | 0.10 (0.02)B | 44 (18)aB | 8 (1)aB | 9.6 (0.2) | 5.4 (2.6)B |
| | CSW2 | 8.2 (1.4) | 0.8 (0.1)ab | 3.49 (0.62) | 0.12 (0.03)B | 26 (7)abB | 8 (2)aB | 10.2 (0.2) | 3.3 (0.4)B |
| Summer | | | | | | | | | |
| | mC | 7.9 (0.8) | 0.8 (0.1)ab | 5.58 (0.67)c | 0.08 (0.02)A | 35 (4)bB | 18 (1)bA | 10.2 (0.4) | 2.0 (0.1)C |
| | CS | 7.6 (0.9) | 0.8 (0.1)ab | 9.47 (1.96)b | 0.08 (0.01)A | 32 (4)abB | 33 (7)bA | 9.8 (0.1) | 1.0 (0.1)C |
| | CSW | 7.6 (0.7) | 0.8 (0.0)b | 7.76 (0.75)b | 0.08 (0.01)A | 43 (7)abB | 28 (4)abA | 9.7 (0.3) | 1.6 (0.3)C |
| | CSW1 | 8.1 (0.8) | 0.9 (0.1)a | 16.68 (0.87)a | 0.37 (0.22)A | 88 (32)aB | 76 (8)aA | 9.0 (0.2) | 1.2 (0.4)C |
| | CSW2 | 8.7 (1.1) | 0.9 (0.1)a | 12.14 (4.03)ab | 0.34 (0.12)A | 54 (7)abB | 68 (13)aA | 9.5 (0.1) | 0.8 (0.1)C |
| Autumn | | | | | | | | | |
| | mC | 8.1 (0.6) | 0.7 (0.1)ab | 1.31 (0.15) | 0.07 (0.02)B | 58 (21)bA | 5 (1)bB | 11.4 (0.3) | 14.3 (7.3)A |
| | CS | 7.7 (1.1) | 0.7 (0.1)ab | 1.44 (0.28) | 0.06 (0.01)B | 46 (15)abA | 5 (1)bB | 10.9 (1.0) | 9.6 (3.2)A |
| | CSW | 7.4 (0.8) | 0.7 (0.1)b | 1.28 (0.30) | 0.08 (0.02)B | 117(77)abA | 6 (2)abB | 10.6 (0.6) | 15.6 (5.2)A |
| | CSW1 | 9.6 (0.6) | 0.9 (0.0)a | 1.41 (0.06) | 0.05 (0.01)B | 102 (27)aA | 7 (1)aB | 10.6 (0.5) | 17.1 (7.2)A |
| | CSW2 | 8.9 (0.9) | 0.9 (0.1)ab | 0.96 (0.15) | 0.05 (0.01)B | 190 (42)abA | 6 (1)aB | 10.4 (0.4) | 30.4 (4.0)A |
| ANOVA Factor | | | | | *P* values | | | | |
| Season | 0.756 | 0.769 | **< 0.001** | **0.004** | **< 0.001** | **< 0.001** | 0.213 | **< 0.001** | |
| Rotation | 0.298 | **0.040** | **< 0.001** | 0.084 | **0.038** | **< 0.001** | 0.223 | 0.947 | |
| Season*Rotation | 0.994 | 0.928 | **< 0.001** | 0.071 | 0.965 | 0.221 | 0.746 | 0.192 | |

Note: Significant comparisons are shown among Rotations (lowercase) and Season (capital) with letters.



Table 2. Soil extracellular enzyme activities (EEA) expressed as nano-moles of product per hour per gram of dry soil.

| Season | Rotation | Soil Extracellular Enzyme Activity (nmol hr$^{-1}$ g$^{-1}$ soil) | | | | | | | |
|--------|----------|-------|--------|-------|--------|---------|--------|--------|--------|
| | | BGase | CBHase | LAPase | NAGase | PHOSase | TAPase | PPOase | PERase |
| **Spring** | | | | | | | | | |
| | mC | 94 (8)b | 27 (2)b | 24 (4)bA | 27 (2)ab | 133 (19)bC | 10 (1)abA | 140 (47)B | 614 (12)a |
| | CS | 107 (18)b | 28 (5)b | 28 (4)abA | 20 (2)b | 129 (20)bC | 11 (0)abA | 100 (30)B | 634 (53)a |
| | CSW | 118 (12)ab | 31 (4)ab | 26 (8)abA | 33 (2)ab | 152 (7)abC | 12 (2)bA | 92 (27)B | 602 (59)ab |
| | CSW1 | 148 (5)a | 50 (5)a | 43 (5)abA | 47 (3)a | 188 (17)aC | 16 (1)aA | 87 (13)B | 516 (24)b |
| | CSW2 | 153(13)ab | 56 (12)ab | 33 (5)aA | 48 (5)a | 208 (8)aC | 16 (1)aA | 137 (61)B | 562 (24)b |
| **Summer** | | | | | | | | | |
| | mC | 100 (5)b | 37 (3)b | 7 (2)bB | 43 (4) | 270 (42)bA | 9 (2)abB | 174 (67)B | 676 (88)a |
| | CS | 111 (17)b | 43 (10)b | 14 (3)abB | 44 (7) | 291 (25)bA | 9 (1)abB | 140 (50)B | 580 (124)b |
| | CSW | 102 (7)ab | 47 (12)ab | 14 (2)abB | 47 (3) | 280 (13)abA | 7 (2)bB | 96 (29)B | 578 (68)b |
| | CSW1 | 146 (12)a | 61 (10)a | 20 (3)abB | 69 (10) | 370 (45)aA | 14 (1)aB | 236 (91)B | 317 (144)bc |
| | CSW2 | 132 (17)ab | 62 (14)ab | 13 (4)aB | 59 (9) | 400 (56)aA | 12 (1)aB | 126 (73)B | 392 (97)c |
| **Autum** | | | | | | | | | |
| | mC | 111 (9)b | 44 (6)b | 5 (3)bB | 67 (13) | 238 (57)bB | 14 (3)abA | 330 (77)A | 543 (113)a |
| | CS | 110 (17)b | 42 (8)b | 8 (1)abB | 55 (7) | 209 (36)bB | 11 (2)abA | 234 (64)A | 461 (103)bc |
| | CSW | 115 (19)ab | 49 (15)ab | 9 (2)abB | 54 (9) | 245 (34)abB | 14 (2)bA | 176 (18)A | 517 (150)b |
| | CSW1 | 138 (10)a | 59 (6)a | 8 (1)abB | 63 (13) | 277 (42)aB | 18 (2)aA | 300 (30)A | 396 (76)c |
| | CSW2 | 117 (15)ab | 46 (8)ab | 17 (3)aB | 63 (2) | 308 (24)aB | 18 (2)aA | 202 (51)A | 336 (49)c |
| ANOVA Factor | | | | | *P* values | | | | |
| | Season | 0.775 | 0.063 | **<0.0001** | **<0.0001** | **<0.0001** | **0.003** | **<0.0001** | **<0.0001** |
| | Rotation | **0.017** | **0.006** | **0.007** | **<0.0001** | **0.0003** | **0.002** | 0.224 | **<0.0001** |
| | Season*Rotation | 0.852 | 0.839 | 0.314 | **<0.0001** | 0.967 | 0.647 | 0.837 | **<0.0001** |

Note: Significant comparisons are shown among Rotations (lowercase) and Season (capital) with letters.



Table 3.  Analysis of variance of results from the principal components analysis of community-level physiological profile (Fig. 4).

| ANOVA§ Parameter | PC1 | | PC2 | | PC3 | | PC4 | | PC5 | | MANOVA (Total) | |
|---|---|---|---|---|---|---|---|---|---|---|---|---|
| Proportion of variance | 38.7 | | 17.7 | | 14.5 | | 9 | | 3.8 | | 83.7 | |
| ANOVA Factor | F | *P* value | F | *P* value | F | *P* value | F | *P* value | F | *P* value | F | *P* value |
| Season | **64.02** | **< 0.001** | **22.57** | **< 0.001** | **5.4** | **0.008** | 0.68 | 0.510 | **10.33** | **< 0.001** | **33.28** | **< 0.001** |
| Crop Rotation | 0.69 | 0.605 | **3.03** | **0.028** | **12.82** | **< 0.001** | 0.36 | 0.834 | 1.81 | 0.146 | **2.19** | **0.003** |
| Season*Rotation | 0.16 | 0.995 | 1.22 | 0.311 | 0.55 | 0.81 | 0.88 | 0.544 | 0.27 | 0.973 | 0.65 | 0.949 |
| Significant comparisons¥ | $1=3\neq2$ | | $1=2\neq3$, CS $\neq$ CSW1 | | $1=2\neq3$, mC=CS$\neq$CSW= CSW2 | | | | $1\neq2=3$, | | | |

§ Degrees of freedom: Season = 2, Crop Rotation = 4, Season*Rotation = 8.

¥ Significant comparison abbreviations: 1 = spring, 2 = summer, 3 = autumn





Table 4. Catabolic evenness by season and crop rotation (showing full suite of C substrates, without carboxylic acids, and carboxylic acids only).

| Season | Rotation | Catabolic Evenness | | |
| --- | --- | --- | --- | --- |
| | | Full | No Carboxylic Acids | Carboxylic Acids Only |
| Spring | | | | |
| | mC | 24.37 (0.79)A | 20.20 (0.05)aA | 7.60 (0.23)aB |
| | CS | 23.79 (0.91)A | 19.80 (0.15)aA | 7.21 (0.13)abB |
| | CSW | 22.98 (0.63)A | 19.65 (0.15)bA | 6.56 (0.35)bB |
| | CSW1 | 24.28 (0.44)A | 18.95 (0.19)abA | 6.91 (0.12)abB |
| | CSW2 | 24.52 (0.72)A | 19.75 (0.24)bA | 6.90 (0.31)bB |
| Summer | | | | |
| | mC | 14.99 (1.61)B | 18.95 (0.59)aA | 4.91 (0.54)aC |
| | CS | 12.86 (1.77)B | 20.20 (0.18)aA | 4.32 (0.38)abC |
| | CSW | 12.10 (1.02)B | 19.82 (0.54)bA | 3.93 (0.20)bC |
| | CSW1 | 13.83 (1.65)B | 18.59 (0.83)abA | 4.34 (0.50)abC |
| | CSW2 | 12.78 (0.92)B | 19.24 (0.51)bA | 3.75 (0.11)bC |
| Autumn | | | | |
| | mC | 25.81 (0.79)A | 19.62 (0.16)aB | 8.47 (0.24)aA |
| | CS | 25.82 (0.55)A | 19.11 (0.22)aB | 8.41 (0.22)abA |
| | CSW | 25.71 (0.74)A | 18.98 (0.28)bB | 8.12 (0.61)bA |
| | CSW1 | 27.41 (0.63)A | 18.63 (0.12)abB | 8.90 (0.24)abA |
| | CSW2 | 26.08 (0.67)A | 18.17 (0.28)bB | 8.11 (0.08)bA |
| | ANOVA Factor | | | |
| | Season | **< 0.001** | **0.002** | **< 0.001** |
| | Treatment | 0.357 | **0.035** | **0.028** |
| | Season*Treatment | 0.928 | 0.058 | 0.807 |

Note: Significant comparisons are shown among Rotations (lowercase) and Season (capital) with letters.



Table 5. Pearson correlation coefficients between soil properties and community-level physiological profile (CLPP) parameters.

| Soil Variable | Substrate Guilds | | | | | Catabolic Evenness | | |
| | Amino acids | Amine | Carboxylic Acids | Carbohydrates | Complex | Full | No Carboxylic Acids | Only Carboxylic Acids |
|---|---|---|---|---|---|---|---|---|
| Water content | ns | ns | ns | ns | ns | 0.40 | ns | **0.52** |
| pH | 0.27 | **0.43** | **-0.41** | ns | **0.53** | **0.68** | ns | **0.74** |
| Sand | **-0.36** | ns | 0.28 | -0.27 | ns | ns | ns | ns |
| Silt | 0.30 | ns | ns | ns | ns | ns | ns | ns |
| Clay | ns | ns | ns | ns | ns | ns | -0.33 | ns |
| Total C | ns | ns | ns | ns | ns | ns | **-0.40** | ns |
| Total N | ns | ns | ns | ns | ns | ns | **-0.40** | ns |
| C-to-N ratio | ns | 0.27 | ns | ns | 0.30 | **0.45** | ns | **0.53** |
| $NH_4^+$ | ns | -0.31 | 0.33 | ns | **-0.37** | **-0.40** | ns | **-0.38** |
| $NO_3^-$ | **-0.58** | **-0.55** | **0.66** | -0.30 | **-0.72** | **-0.74** | ns | **-0.70** |
| PMC | ns | 0.29 | ns | ns | ns | ns | **-0.63** | ns |
| PMN | ns | -0.27 | 0.32 | ns | **-0.55** | **-0.49** | ns | **-0.52** |
| MBC | 0.31 | **0.49** | **-0.37** | ns | ns | **0.41** | **-0.38** | **0.47** |
| MBN | **0.36** | 0.34 | **-0.37** | **0.42** | ns | **0.36** | ns | 0.31 |
| MBC:MBN | ns | **0.40** | ns | ns | ns | 0.31 | **-0.34** | **0.40** |
| BGase | ns | **-0.43** | **0.30** | ns | ns | -0.29 | 0.32 | -0.28 |
| CBHase | -0.32 | **-0.47** | **0.39** | -0.27 | ns | -0.33 | ns | -0.28 |
| LAPase | ns | -0.29 | ns | ns | ns | ns | **0.49** | ns |
| TAPase | ns | **-0.37** | ns | ns | ns | ns | ns | 0.37 |
| NAGase | **-0.35** | **-0.56** | **0.47** | **-0.39** | -0.29 | **-0.46** | **0.29** | **-0.41** |
| PHOSase | **-0.45** | **-0.66** | **0.56** | **-0.46** | -0.34 | **-0.63** | **0.34** | **-0.60** |
| PPOase | -0.38 | -0.33 | 0.37 | -.31 | ns | ns | ns | ns |
| PERase | -0.40 | -0.54 | 0.42 | -0.37 | ns | -0.30 | 0.43 | ns |

Note: Only significant correlations are shown (*P* values < 0.05), ns = non-significant

**FIGURES**

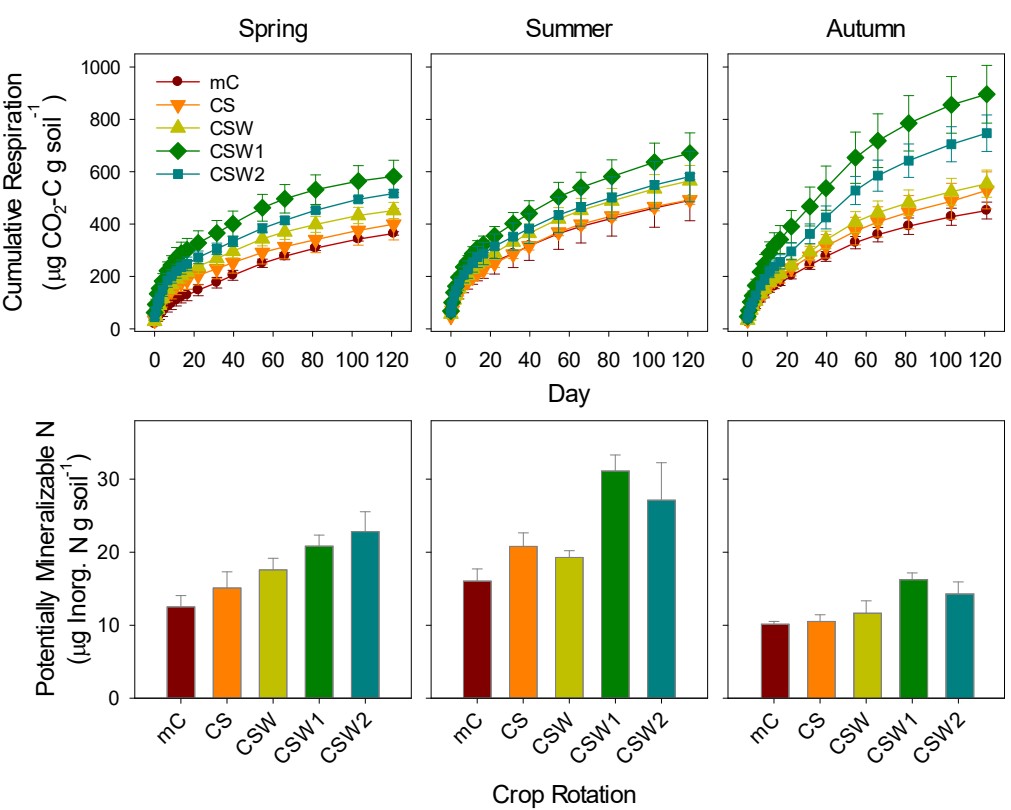

Figure 1.  Potentially mineralizable carbon (top  row) and potentially mineralizable nitrogen
(bottom row).  Crop rotation abbreviations are: monoculture corn (mC), corn-soy (CS), corn-soy-
wheat (CSW), corn-soy-wheat with red clover cover crop (CSW1), and corn-soy-wheat with red
clover + rye cover crops (CSW2).  Means are shown and error bars are standard errors (n = 4).




Figure 2. Soil microbial biomass parameters by season and crop rotation. See Fig.1 for crop

rotation abbreviations.  Means are shown and error bars are standard errors (n = 4).



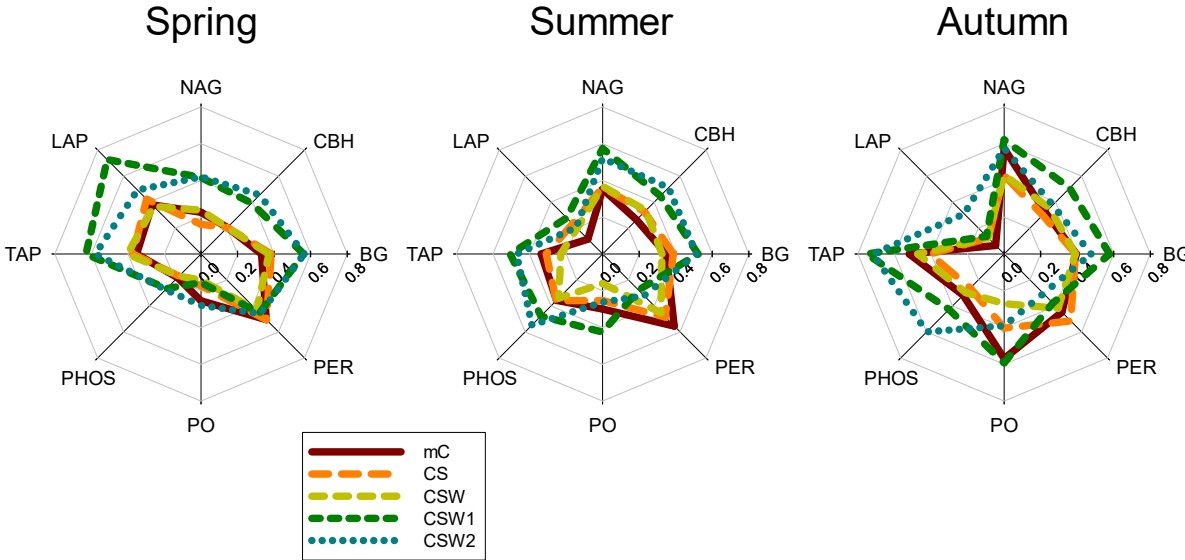

Figure 3. Extracellular enzyme activities (EEA) normalized for the maximum value during each season. EEA abbreviations are: β-1,4,-glucosidase (BG), β-D-1,4-cellobiohydrolase (CBH), β-1,4,-N-acetyl glucosaminidase (NAG), acid phosphatase (PHOS), Tyrosine aminopeptidase (TAP), Leucine aminopeptidase (LAP), phenol oxidase (PO), and peroxidase (PER). See Fig.1 for crop rotation abbreviations.




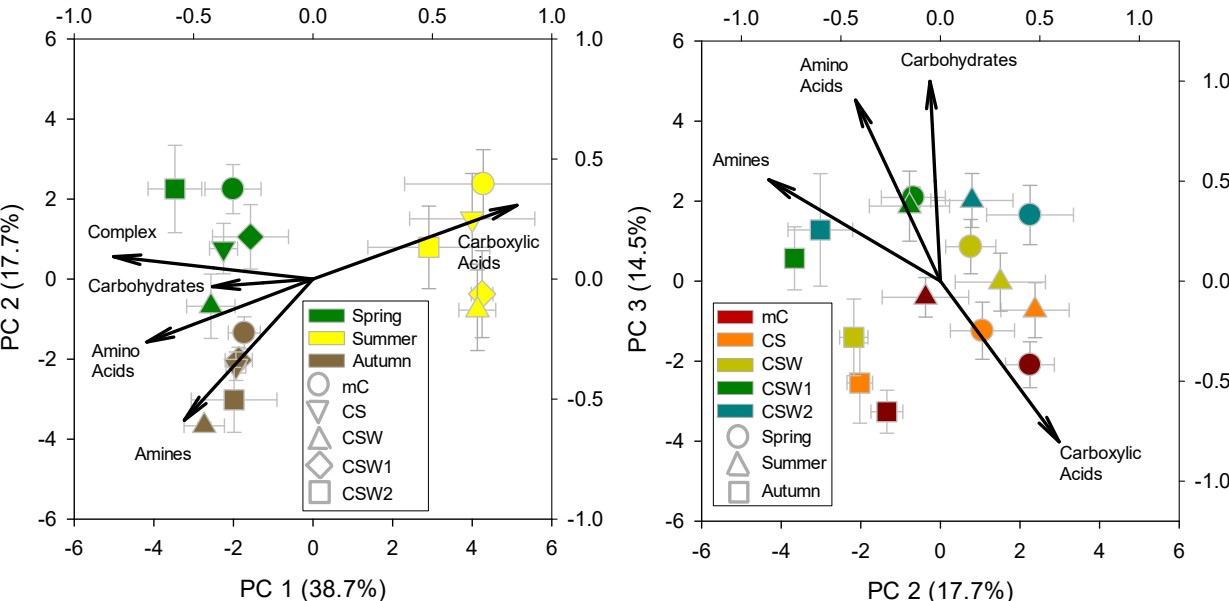

Figure 4. Principal components analysis (PCA) on all 31 substrates. *Left Panel:* Principal components 1 and 2, where Season is

dominant discriminating factor ($P < 0.001$) and *Right Panel:* Principal components 2 and 3 where Rotation is highlighted as a

dominant discriminating factor. See also Table 5 for PCA and ANOVA results. Means are shown and error bars are standard errors

(n = 4). See Fig.1 for crop rotation abbreviations.