# Peer review of "Soil microbial biomass and function are altered by 12 years of crop rotation"

_SOIL, 2016_

## Referee Comment (RC1) · Anonymous Referee #1 · 21 Jul 2016

I acknowledge the amount of work put into this study by the authors. This is a unique and comprehensive investigation of the effects of a single management practice (rotation) on microbial function, where many studies cannot study individual management decisions due to other, confounding management decisions. The introduction and discussion sections are very clear, with minimal jargon, and the data interpretation is logical. However, some issues need to be addressed (some very minor, others more crucial):

1. Throughout the manuscript: I noticed several problems with regards to the references, e.g. wrong year of publication reported in the main body of the text (Treseder et al. 2015; Anderson & Domsch, 1989), missing citations in the reference section (Ret et al. 2008; Guckert et al., 1996; Paul et al., 1999; Robertson et al., 2000; Schimel et al., 1989; Tibbits et al., 2002; Mou et al., 2008; Frey et al., 1999), references not cited in the

main body of the text (Adviento-Borbe et al. 2010; Behnke-Ryser et al., 2012; Berard et al., 2015; Cambardella et al., 1999; Morillas et al., 2015; Plante et al., 2011; Thoms et al., 2010; Trivedi et al., 2015), and authorship misreported (e.g. "McDaniel (2014c)" instead of "McDaniel et al. (2014c)"; "Giller, K. E. N. E." instead of "Giller, Ken E."; "Van Der Putten, W. I. M. H." instead of "Van Der Putten, Wim H."). References were also not in alphabetical order in some instances. Additionally, there were spelling mistakes for some of them (see Doran et al., 2000; Franzluebbers et al., 1995; Hamilton et al., 2015), and incomplete references for Venter et al., 2016. Please make appropriate changes where needed. 2. L. 31: The authors may want to reconsider the statement as species richness in a rotated cropping system is one only if all weeds have been removed from the system, which is theoretically possible but not always the case. 3. L. 52-64: The authors introduce CLPP and how it works. Being unfamiliar with these measurements and how to interpret them, I think it would be helpful to other readers like myself if the authors added more information about this technique in the context of agricultural studies. How would a more or less even CLPP profile be interpreted in the context of agricultural soils? Does an even profile generally correlates with more efficient nutrient transfer to the plant or perhaps better C retention? Are there any previous studies that have looked into this that can be referred to here? 4. L. 83: 42° 24' N? 5. L. 96 & 98: It is Zea mays and Trifolium pratense. Please correct. 6. L. 101: Where in the field were the soil cores taken during the summer? Were they within or between rows? Likewise, were cores in the spring and autumn timepoints taken where previous rows or interrows had been? The rhizosphere effect can have a large impact on microbial communities and functioning and the summer collection is the only time point in which corn is actively growing. This might change interpretation of the data. 7. L. 114-122: The amount of sample used for specific measurements is omitted and should be included for reproducibility. Also, why specifically using a 50% water-holding capacity? 8. L. 127-129: What is the rationale for measuring PMC more frequently at the beginning of the experiment than the end? 9. L. 133: The acronyms MBC and MBN should be introduced here for later use in the manuscript. 10. L. 142: "Soils were

analyzed for 7 extracellular enzyme activities" should be changed to "8". 11. L. 173: Please indicate what sort of transformation was applied to the data (remind the reader in the Figure titles as well). 12. L. 184: it is conventionally accepted to provide details about the version of R used. Later versions generally have bugs fixed and may explain discrepancies observed by other users (if they were to run the exact same dataset as the authors'). 13. L. 195: "There were no significant rotation or season effects on total soil C and N". Table 1 says otherwise. Please correct and, if necessary, adapt your interpretation of the data and conclusions. 14. L. 198-212: Clearer language should be used throughout the results section. In several instances the authors average over the rotation treatments or the season treatments without telling the reader, making the percentage increases difficult to interpret (see lines 199, 200, 202, 204, 206-208 [why combining CSW and CS treatments here?]). Furthermore, I find different results for the DOC:DON mean in autumn (i.e. 17.4, five times that of spring and 13 times that of the summer). Report the standard deviation or standard error for these calculations. 15. L. 217-218: This is impossible to state without a post-hoc test. Please use the appropriate test and rephrase if necessary. 16. L. 221-222: I disagree. These effects seem strongest in the spring and autumn. 17. L. 228: This is not the p-value reported in the Figure. 18. L. 240-241: It should be "25% greater than autumn and 99% greater than spring". 19. L. 256 and 272: I do not understand how the authors obtained these values. Could Table 4 be erroneous? 20. L. 263: Refer to table 4 instead of Figure S3. 21. L. 274: "Complementary" not "complimentary". 22. L. 279: I agree that there is a positive relationship between CLPP and EEA but I would not say they are "strong". 23. L.296-307: This is almost the same paragraph as l.273-295. Please remove. 24. L. 339-359: The authors discuss how drying and wetting impacted their results in the summer treatment. Why was this date chosen as a sampling date? Was the rain just bad luck? The fact that other timepoints were during dry periods may confound the comparison between summer and other seasons. The study may have benefited from another summer timepoint taken when a wetting period had not occurred for comparison. The author's comments on this would be appreciated. 25. L. 347: It should be

"Table 1", not "Table 2". 26. L. 365: Are the authors sure about these values? I was unable to find the same results. 27. L. 376: The authors refer to microbial biomass C:N as having a season × rotation interaction. However, Table 4 does not show this interaction as statistically significant. Is the data in the table incorrect or is the text wrong? In the case that microbial biomass C:N does not have this interaction does the following interpretation (line 377-378) that these interactions are "indicative of the enhanced ability of soil microbes under diverse rotation to process, provision, and retain N" still hold? 28. L. 376-378: Regardless of what measurement showed season × rotation interaction, the interpretation that these interactions serve as evidence for enhanced N cycling and retention in diverse systems could use strengthening. What specific functions of NAG and PER indicate that they can improve N cycling, and why is an interaction between season × rotation meaningful? If microbial biomass C:N does show an interaction, how specifically does this serve as evidence for enhanced N provisioning? As it stands, these lines seem like a very important argument in the manuscript with little discussion to strengthen it. 29. L. 405: Repetition of "studies". 30. L. 471-472: The authors may want to reconsider the strength of their statement. There is no proof rotation facilitate microbes in supplying more N to crops; there is only more potentially mineralizable N thanks to diversification (and the word "potentially" is important in this context). 31. Tables: 1-page Tables would be much more convenient to read and the number in parentheses/use of bold text/colors need to be explained in the title. Additionally, Tables S2, S3, and S4 are not reported in the text. 32. Figures: I would have liked to find post-hoc analyses for each panel in Figure 1 and 2. The statistics used for Figure 2 need to be briefly explained in the title, as well as what the error bars represent in Figure S2. The legend for Figure S3 is too vague; what are the lines and icons representing? Finally, there is an overall problem with Figure numbering in the Supplementary Material.

---

## Referee Comment (RC2) · Anonymous Referee #2 · 22 Jul 2016

General comments

The authors examine whether using crop rotations to increase temporal biodiversity within an agroecosystem enhances soil biochemical functioning. Specifically, they hypothesised that crop rotations will enhance catabolic diversity (through community-level physiological profiles) and soil function (enzyme activities, soil microbial biomass, potentially mineralizable C and N). Further, they hypothesized that the crop rotation effect would lessen over the growing season. The study used soils from a well-established at the W.K. Kellogg Biological Station (est. 2000).

I find the paper well-structured and easy to follow with important findings that contribute new knowledge on the effect of crop rotation and soil biology and function. The study is quite unique in that there are few management variables (fertilization, pest control

etc.) that could confound the effect of crop rotation.

The introduction appears to have an unbalanced focus on CLPP and soil substrate use, while neglecting research gaps/other studies relating to extracellular enzyme activities.

The methods appear valid and adequate to test the hypotheses.

Results are well communicated, although supplementary data are disorganised and do not link to the present manuscript. Serious repetition of sentences from L273-285 in L296-308.

Discussions and conclusions are well-substantiated by the results, although some aspects relating to cover crops may enhance the discussion further – see specific comments below. Further, there is a lack of discussion around the enzyme activities (as was the case in the introduction).

There are also a few referencing issues with some references being cited in the text and not listed in the reference list and visa versa.

Specific comments

L1 – I don't find any reference to catabolic evenness or diversity in the Abstract; a main component of your first hypothesis. L36 – replace "of" with "on" L44-51 – The end of this paragraph does not seem to be relevant to identifying gaps in the knowledge around above- and belowground biodiversity relationships (the point raised at the beginning of the paragraph). Perhaps you are expanding on the link to ecosystem function? Then I would suggest a new paragraph dealing with this. L56-67 – Unclear whether "their" refers to "soil microbial functions" or to "crop rotations". Re-structure to make it clearer. L85 – hyperlink takes you to a page that no longer exists L86 – Was one crop planted per year, or multiple within one year? I know this might be obvious, but in some rotation systems, there are multiple plantings per year. L98-99 - When and how were cover crops (in CSW1 and CSW2) planted and was an entire growing season dedicated to this? - i.e. was it a 4-year rotation or a 3-year rotation with cover

crop grown in between corn, soy and wheat cropping dates. L146 – Would freezing of the samples for EEA analysis deplete the absolute enzyme activity? – perhaps substantiate this with references to other studies that have done likewise. L165 – Why are only two readings taken and why after only 6 hours? Does the $CO_2$ efflux plateau within 6h? L241 – "season had no. . .." the first half of this sentence is a bit clumsy and difficult to understand. Perhaps re-word it. L273-285 – I don't see how the correlation between EEA and CLPP contributes to the overall thesis of the study. These results do corroborate each other and evidence the reliability of the CLPP and EEA data, but, in my opinion do not warrant such a long paragraph in the Results, especially since there is no follow-up discussion points in the Discussion section. I would advise simplifying or leaving this out. L276 – remove "quite" L278 – The Fig. S4 does not relate to Nag amine. I think the order of supplementary figures is incorrect and does not correspond to the manuscript. Please check this throughout. L296-308 – all this is a repetition of L273-285. Remove either section and simplify as suggested above. L309 – I do not see any discussion around cover crops and how they affect soil biochemical responses relative to non-cover crop treatments. Increases in soil biochemical functioning may not be a result of plant species diversity per se - rather, cover crops alter soil physical characteristics (e.g. soil moisture through covering soil in between cash crops) which drive changes in biochemical processes. I would suggest clarifying the definition of cover crops in the methods and expanding on their relative effects on soil physico-chemical characteristics in the discussion. L321 – I do not see any direct reference to the second hypothesis here. It will make it easier for the reader to follow if this is done (as you have done in L393). L336 – I would advise some discussion (also in relation to the cover crops) about legumes and soil N. Increases in microbial biomass may not be driving increased N, rather key-stone microbial species (rhizobia) may be responsible. L342 – what are the units for "0.1"? L372 – Again, I would suggest making the link to the original hypothesis more explicit (partly done in L396 but would suggest doing this earlier as well). L416-419 – This is a confusing sentence, please re-word. L427-431 – I do not understand this logic. Do you mean to say that using CLPP as a

measure of catabolic evenness in bacterial-dominated soils may not adequately reflect the true microbial catabolic diversity because (1) bacteria are generalists and use all substrates evenly, and (2) fungi tend to be excluded through disturbing the soil? If so please re-structure this or explain what you are attempting to say. L431-433 – How does this support the previous statements? 16S rRNA diversity would not necessarily correspond to catabolic evenness, so cannot be used to firmly support your findings. Table 1. – Include full stop. To which variables do the units apply to? (e.g. does mg.kg-1 apply to C:N ratio?)  - make clearer please.  Give full descriptions of crop rotation abbreviations in the title.

---

## Referee Comment (RC3) · Anonymous Referee #3 · 22 Jul 2016

**Soil-2016-39 Review comments**

**General comments**:

The article has investigated the effects of temporal plant diversification by rotating crop on soil microbial functions and activities, while many other studies focus on the above-ground spatial biodiversity on soil microbial functions. This study has comprehensively examined a suite of indexes of soil microbial activities (e.g., MBC, MBN, PMC, and PMN) and functions (e.g., EEA and CLPP) over one growing season. The introduction is clear, the methods are easy to follow although they need clarification, and the data interpretation is generally logic. But a couple of issues need to be addressed as follows:

**Majors**:

1)  The main goal of this study was to investigate whether crop rotation can enhance soil microbial biomass and functions. Since the study has examine microbial biomass and functions (e.g., PMC, PMN, and EEA) over one growing season (i.e., in spring, summer, and fall), how confidently we can attribute enhanced microbial activities and functions to crop rotation rather than seasonality or their interactions? Can we confidently say that crop rotation is the main reason for changes in soil microbial functions and activities?
2)  Were the crop rotation effects on microbial activities and functions general or unique, considering that all the measurements were for soils that have experienced an extreme drought the year before sampling? What the results will be if the sampling took place in a normal year? Particularly, the authors have discussed that the drying-rewetting effects on soil microbial community in the discussion (L339-359).

**Minors**:

1)  Problems with the reference order in the main text: L27-28, L41-42, L50-51, L120, L157, 380, 381, 385-386, 398-399, 401-402, 423
2)  L54-58, it is better to introduce the rationale of CLPP method and substrates used in this method, especially for readers who are not familiar with them. More references are needed about how this method works and how widely it has been used in studies to examine soil microbial catabolic functions.
3)  Soil sampling issue. I am confused whether or not the measured soils were sampled under the same crop (maize) in all the three seasons. According the statement in L99-101, it is not the case. Soil sampling took place in April, July, and November, while corn was planted in June and before June some plots may be planted with other crops. Please clarify.
4)  PMC and PMN measurements. How long did the incubation last? It is 6 months in line 122, but it is 120 days in line 129. How was PMC calculated according to cumulative respiration? In addition, how many times were the inorganic N extraction conducted to assess PMN during the entire period of incubation? And how was PMN calculated and what is the reference for it? I cannot find it (L117-131).
5)  L198, 375 please use the multiple symbol instead of "X" letter.
6)  Line 2013 should it be "P" rather than "Ps"?
7)  L210-211, please clarify the relation between cumulative $CO_2$ respiration and PMC. Otherwise, the statement in these lines is not true.

8) L228-230, the text "increased crop diversity decreasing the qCO2 by 16, 40, 28% in CSW, CSW1, and CSW2…' does not match the results in Fig. 2. In CSW, qCO2 was not always decreased compared to mC (i.e., the control) according to Fig. 2 in spring, summer, and fall. Moreover, is the difference statistically significant?

9) L291 "negative" should be "negatively".

10) In the discussion, I would like to read the discussion of "crop diversity and soil microbial functions" before "seasonal dynamics and N limitation", as the former is the focus of this study, and they directly answer the two questions asked in the introduction.

11) L448-453 the statement does not match Figure. S4. We cannot tell which data are from the less and which are from the more crop rotations from this Figure. Please clarify.

12) The statements in L399-400 and L411-412 are not clear to me. Which tables or figures can show the results? And what are the indexes of microbial catabolic evenness?

13) L471-472 as a main finding of this study, it should be discussed in details but it was not. Please specify or remove it.

**Figures and tables**

Tables 1 and 2: it is easy to read when all the contents of a table are in one page.

Table 3 caption. I do not understand why adding Fig. 5 at the end? What does it mean? This is no Fig. 5 in the manuscript.

Fig. 1: as mentioned before, cumulative respiration is different from potential mineralizable C, please clarify the relation between them.

Figs. 1 and 2: since seasonality has strong effects on soil microbial activity and functions, we need to know how crop rotation effects in each season. It is better to add the multiple comparison of soil microbial activity in each season in these two figures.

---

## Author Comment (AC1) · 8 Sep 2016

Anonymous Referee #1

Author's responses in blue.

I acknowledge the amount of work put into this study by the authors. This is a unique and
comprehensive investigation of the effects of a single management practice (rotation) on microbial
function, where many studies cannot study individual management decisions due to other, confounding
management decisions. The introduction and discussion sections are very clear, with minimal jargon,
and the data interpretation is logical. However, some issues need to be addressed (some very minor,
others more crucial):

Response: Thank you for this comprehensive and thorough review.  We really appreciate the time and
effort this reviewer put in on their feedback.  We have incorporated nearly all their suggestions and we
feel it has greatly improved the manuscript.

1. Throughout the manuscript: I noticed several problems with regards to the references, e.g. wrong
year of publication reported in the main body of the text (Treseder et al. 2015; Anderson & Domsch,
1989), missing citations in the reference section (Ret et al. 2008; Guckert et al., 1996; Paul et al., 1999;
Robertson et al., 2000; Schimel et al., 1989; Tibbits et al., 2002; Mou et al., 2008; Frey et al., 1999),
references not cited in the main body of the text (Adviento-Borbe et al. 2010; Behnke-Ryser et al., 2012;
Berard et al., 2015; Cambardella et al., 1999; Morillas et al., 2015; Plante et al., 2011; Thoms et al., 2010;
Trivedi et al., 2015), and authorship misreported (e.g. "McDaniel (2014c)" instead of "McDaniel et al.
(2014c)"; "Giller, K. E. N. E." instead of "Giller, Ken E."; "Van Der Putten, W. I. M. H." instead of "Van Der
Putten, Wim H."). References were also not in alphabetical order in some instances. Additionally, there
were spelling mistakes for some of them (see Doran et al., 2000; Franzluebbers et al., 1995; Hamilton et
al., 2015), and incomplete references for Venter et al., 2016. Please make appropriate changes where
needed.

Response: Thank you for this thorough reporting of the reference mistakes, we appreciate it!  I
apologize for you having to spend the time finding these mistakes.  I am quite embarrassed by all of
these errors, and learned you should not rely on your citation manager program.  The references have
now been thoroughly checked.

2. L. 31: The authors may want to reconsider the statement as species richness in a rotated cropping
system is one only if all weeds have been removed from the system, which is theoretically possible but
not always the case.

Response: This is true, we have modified the statement to include the consideration of weeds (L. 34).
Thank you for this observation.

3. L. 52-64: The authors introduce CLPP and how it works. Being unfamiliar with these measurements
and how to interpret them, I think it would be helpful to other readers like myself if the authors added
more information about this technique in the context of agricultural studies. How would a more or less
even CLPP profile be interpreted in the context of agricultural soils? Does an even profile generally
correlates with more efficient nutrient transfer to the plant or perhaps better C retention? Are there any
previous studies that have looked into this that can be referred to here?

Response: Good point, we have now illustrated how CLPP data could be useful in an agricultural context
(L. 63-78).

4. L. 83: 42◦ 24' N?

Response: This has been corrected.

5. L. 96 & 98: It is Zea mays and Trifolium pratense. Please correct.

Response: These have been corrected.

6. L. 101: Where in the field were the soil cores taken during the summer? Were they within or between
rows? Likewise, were cores in the spring and autumn timepoints taken where previous rows or
interrows had been? The rhizosphere effect can have a large impact on microbial communities and
functioning and the summer collection is the only time point in which corn is actively growing. This
might change interpretation of the data.

Response: We have indicated where the soil cores were collected (between the rows, L. 118).

7. L. 114-122: The amount of sample used for specific measurements is omitted and should be included
for reproducibility. Also, why specifically using a 50% water-holding capacity?

Response: We indicated the amount of soil in the incubation (10 g, L. 137), and that 50% WHC was used
because it is near optimal water content for respiration in these soils (L. 138).

8. L. 127-129: What is the rationale for measuring PMC more frequently at the beginning of the
experiment than the end?

Response: The reason for higher frequency measurements were two-fold: 1) reduce $CO_2$ build-up and
lack of $O_2$ in the jar when respiration rates are extremely high, and 2) to get better resolution of the
exponential portion of the $CO_2$ "decay curve" for modeling C pools (data we did not use in this
manuscript). We state this now in the manuscript (L. 146-147).

9. L. 133: The acronyms MBC and MBN should be introduced here for later use in the manuscript.

Response: We have abbreviated them here (L. 153).

10. L. 142: "Soils were for 7 extracellular enzyme activities" should be changed to "8".

Response: This was changed.

11. L. 173: Please indicate what sort of transformation was applied to the data (remind the reader in the
Figure titles as well).

Response: We have now indicated which variables were transformed, and how they were transformed
(L.199- 201).

12. L. 184: it is conventionally accepted to provide details about the version of R used. Later versions
generally have bugs fixed and may explain discrepancies observed by other users (if they were to run
the exact same dataset as the authors').

Response: We have now included the version of R we used (v3.0.0, L. 208).

13. L. 195: "There were no significant rotation or season effects on total soil C and N". Table 1 says
otherwise. Please correct and, if necessary, adapt your interpretation of the data and conclusions.

Response: We have clarified this statement to reflect there are small differences in total soil C and N (L.
221-223).

14. L. 198-212: Clearer language should be used throughout the results section. In several instances the
authors average over the rotation treatments or the season treatments without telling the reader,
making the percentage increases difficult to interpret (see lines 199, 200, 202, 204, 206-208 [why
combining CSW and CS treatments here?]). Furthermore, I find different results for the DOC:DON mean
in autumn (i.e. 17.4, five times that of spring and 13 times that of the summer). Report the standard
deviation or standard error for these calculations.

Response: Thank you for noticing these, and we appreciate the reviewers comment on clarification.  We
have reported standard error throughout the section where we mention means, and clarified the
language.

15. L. 217-218: This is impossible to state without a post-hoc test. Please use the appropriate test and
rephrase if necessary.

Response: We have provided the post-hoc test $P$ values in the text (L. 258-261), and post-hoc test results
in Figs 1 & 2.

16. L. 221-222: I disagree. These effects seem strongest in the spring and autumn.

Response: This sentence was removed and we only talk about the rotation effect now (L. 254).

17. L. 228: This is not the p-value reported in the Figure.

Response: The one reported in the figure was correct, we fixed the text to match it (L. 261).

18. L. 240-241: It should be "25% greater than autumn and 99% greater than spring".

Response: This has been changed.

19. L. 256 and 272: I do not understand how the authors obtained these values. Could Table 4 be
erroneous?

Response: These are the values we received when calculating catabolic evenness.  We double checked
our calculations and compared with other studies (Degens et al. 2000, 2001; Carney & Matson 2005; Sall
et al. 2015).  Of course it is dependent on how many substrates you use, but our values are in the range
of what has been published in the literature (from 8 to 24).

Degens, B. P., Schipper, L. A., Sparling, G. P., & Vojvodic-Vukovic, M. (2000). Decreases in
organic C reserves in soils can reduce the catabolic diversity of soil microbial communities. *Soil*
*Biology and Biochemistry*, *32*(2), 189-196.

Degens, B. P., Schipper, L. A., Sparling, G. P., & Duncan, L. C. (2001). Is the microbial
community in a soil with reduced catabolic diversity less resistant to stress or disturbance?. *Soil*
*Biology and Biochemistry*, *33*(9), 1143-1153.

Carney, K. M., & Matson, P. A. (2005). Plant communities, soil microorganisms, and soil carbon
cycling: does altering the world belowground matter to ecosystem functioning? *Ecosystems*, *8*(8),
928-940.

Sall, S. N., Ndour, N. Y. B., Diédhiou-Sall, S., Dick, R., & Chotte, J. L. (2015). Microbial response
to salinity stress in a tropical sandy soil amended with native shrub residues or inorganic
fertilizer. *Journal of environmental management*, *161*, 30-37.

20. L. 263: Refer to table 4 instead of Figure S3. 21.

Response: This was changed (L. 301) to Fig. S5 and Tables S2 and S3.  The MANOVA results were not in
Table 4, just the catabolic evenness.

L. 274: "Complementary" not "complimentary".

Response: This was changed (L. 324).

22. L. 279: I agree that there is a positive relationship between CLPP and EEA but I would not say they
are "strong".

Response: We refer to the relationship as "significant" now (L. 328).

23. L.296-307: This is almost the same paragraph as l.273-295. Please remove.

Response: The first paragraph was deleted, and left in the "Relationships…" section.

24. L. 339-359: The authors discuss how drying and wetting impacted their results in the summer
treatment. Why was this date chosen as a sampling date? Was the rain just bad luck? The fact that other
timepoints were during dry periods may confound the comparison between summer and other seasons.
The study may have benefited from another summer timepoint taken when a wetting period had not
occurred for comparison. The author's comments on this would be appreciated.

Response: This reviewer is correct.   In a sense, it was just "bad luck."  Apriori planning dictated our
sampling time, but this is of course confounded with climate conditions.  And dry periods are common in
the summer at our research location.  It would be preferable to have more than 3 samples collected
over the year, but because many of the methods here are very labor intensive (i.e. CLPP and enzymes)
we were limited to three sampling events.  That being said, however, we are the first to our knowledge
to have run community-level physiological profiles on the same soils on 3 dates (one of which includes
this unique dry-wet event).

25. L. 347: It should be "Table 1", not "Table 2".

Response: We have changed this (L. 487).

26. L. 365: Are the authors sure about these values? I was unable to find the same results.

Response: Thank you, we had incorrect values here and have changed these to the correct values (L.
505)

27. L. 376: The authors refer to microbial biomass C:N as having a season × rotation interaction.
However, Table 4 does not show this interaction as statistically significant. Is the data in the table
incorrect or is the text wrong? In the case that microbial biomass C:N does not have this interaction does the following interpretation (line 377-378) that these interactions are "indicative of the enhanced
ability of soil microbes under diverse rotation to process, provision, and retain N" still hold?

Response: Yes, it should still hold.  Our MBC:MBN values did show significant interaction (Fig. 2).  We
also added some further interpretation (and evidence) for how these interactions might be indicators of
this (L. 351-377).

28. L. 376-378: Regardless of what measurement showed season × rotation interaction, the
interpretation that these interactions serve as evidence for enhanced N cycling and retention in diverse
systems could use strengthening. What specific functions of NAG and PER indicate that they can
improve N cycling, and why is an interaction between season × rotation meaningful? If microbial
biomass C:N does show an interaction, how specifically does this serve as evidence for enhanced N
provisioning? As it stands, these lines seem like a very important argument in the manuscript with little
discussion to strengthen it.

Response: While we do not have any direct measurements, such as tracer $^{15}$N data, we do have some
evidence for this.  Probably the strongest evidence is the correlation between potentially mineralizable
net N (PMN) and yield.  We have provided some elaboration on this, further support for enhanced N
cycling, and we reworded as more speculative (L. 351-377).

29. L. 405: Repetition of "studies".

Response: The second "studies" has been removed.

30. L. 471-472: The authors may want to reconsider the strength of their statement. There is no proof
rotation facilitate microbes in supplying more N to crops; there is only more potentially mineralizable N
thanks to diversification (and the word "potentially" is important in this context).

Response: We now state this more speculatively (L. 523-524).  While we do not have direct evidence
that microbes are supplying more soil N to the crops, we have some strong inferential evidence in the
relationship between yield and potentially mineralizable N (Fig. S8).

31. Tables: 1-page Tables would be much more convenient to read and the number in parentheses/use
of bold text/colors need to be explained in the title. Additionally, Tables S2, S3, and S4 are not reported
in the text.

Response: We have made the tables 1-page, and added further explanation to the captions.   Also we
now refer to all Tables in the text, except for Table S4, which we removed.

32. Figures: I would have liked to find post-hoc analyses for each panel in Figure 1 and 2. The statistics
used for Figure 2 need to be briefly explained in the title, as well as what the error bars represent in
Figure S2. The legend for Figure S3 is too vague; what are the lines and icons representing? Finally, there
is an overall problem with Figure numbering in the Supplementary Material.

Response: We have put the post-hoc analyses for the crop rotation in Figs. 1 and 2 – but only for the
crop rotation factor.  We feel this is the factor of most importance, and discussing the season post-hoc
results in the text of the manuscript is sufficient.  Sometimes figures can become too busy with letters
indicating more than one factor at a time.   We have also further explained the statistics in both Fig. 1
and 2.  Furthermore we have explained that the error bars are the errors associated with the PC loading values in Fig. S2.    Also, we have fixed the supplemental figure number, thanks for noticing these things
in the supplemental figures.

---

## Author Comment (AC2) · 8 Sep 2016

Referee #2

Authors' responses are in blue.

General comments

The authors examine whether using crop rotations to increase temporal biodiversity within an
agroecosystem enhances soil biochemical functioning. Specifically, they hypothesised that crop
rotations will enhance catabolic diversity (through community-level physiological profiles) and soil
function (enzyme activities, soil microbial biomass, potentially mineralizable C and N). Further, they
hypothesized that the crop rotation effect would lessen over the growing season. The study used soils
from a well-established at the W.K. Kellogg Biological Station (est. 2000).

I find the paper well-structured and easy to follow with important findings that contribute new
knowledge on the effect of crop rotation and soil biology and function. The study is quite unique in that
there are few management variables (fertilization, pest control etc.) that could confound the effect of
crop rotation.

Response: Thank you for this thorough review and kind comments.  We really appreciate the time and
effort this reviewer put in on their feedback.  We have incorporated nearly all their suggestions and we
feel it has greatly improved the manuscript.

The introduction appears to have an unbalanced focus on CLPP and soil substrate use, while neglecting
research gaps/other studies relating to extracellular enzyme activities.

Response: We would agree with this reviewer's assessment, but we have done this intentionally.  The
CLPP is the most novel aspect of the manuscript.  And to our knowledge, no one has done this over
multiple dates from the same soils.  That being said, we still have added some more in the Introduction
and Discussion regarding the extracellular enzymes (L 54-55, 366-368, 503).

The methods appear valid and adequate to test the hypotheses.

Response: Thank you.

Results are well communicated, although supplementary data are disorganised and do not link to the
present manuscript. Serious repetition of sentences from L273-285 in L296-308.

Response: We have removed this duplication and link all supplementary data directly to the manuscript.

Discussions and conclusions are well-substantiated by the results, although some aspects relating to
cover crops may enhance the discussion further – see specific comments below. Further, there is a lack
of discussion around the enzyme activities (as was the case in the introduction).

Response: We have now added more on the importance of the cover crop treatments (L. 378-392) and
some more about the extracellular enzymes (L 366-368, 465-478, 503).

There are also a few referencing issues with some references being cited in the text and not listed in the
reference list and visa versa.

Response: We have fixed these references, and have thoroughly checked the citations and reference
section.

Specific comments

L1 – I don't find any reference to catabolic evenness or diversity in the Abstract; a main component of
your first hypothesis.

Response: We have now added catabolic evenness in the abstract (L. 10, 17).

L36 – replace "of" with "on"

Response: This was changed.

L44-51 – The end of this paragraph does not seem to be relevant to identifying gaps in the knowledge
around above- and belowground biodiversity relationships (the point raised at the beginning of the
paragraph). Perhaps you are expanding on the link to ecosystem function? Then I would suggest a new
paragraph dealing with this.

Response: Good suggestion, this was changed to a new paragraph and more details and context about
CLPP in agroecosystems (L. 68-78).

L56-67 – Unclear whether "their" refers to "soil microbial functions" or to "crop rotations". Re-structure
to make it clearer.

Response: We have replaced "their" with "rotation" to make it clearer (L. 80).

L85 – hyperlink takes you to a page that no longer exists

Response: The hyperlink was changed, and now works.  Thank you for checking this.

L86 – Was one crop planted per year, or multiple within one year? I know this might be obvious, but in
some rotation systems, there are multiple plantings per year.

Response: For most of the year there was just one crop at a time, but there was actually some overlap at
the end of the growing season when red clover was inter-seeded in CSW1 and CSW2 treatments (now
clarified in L. 113-115).

L98-99 - When and how were cover crops (in CSW1 and CSW2) planted and was an entire growing
season dedicated to this? - i.e. was it a 4-year rotation or a 3-year rotation with cover crop grown in
between corn, soy and wheat cropping dates.

Response: See previous response for the answer (L. 113-115).  We have also added another
supplementary figure for further clarification (Fig. S1).

L146 – Would freezing of the samples for EEA analysis deplete the absolute enzyme activity? – perhaps
substantiate this with references to other studies that have done likewise.

Response: Freezing has been shown to slightly decrease absolute activity in some studies (Peoples and
Koide 2012), and no effect in others (Lee et al. 2007, Deforest, 2009).  While we would like our EEA
measurements to be as accurate as possible, we are mostly concerned with relative differences among
treatments.  Thus, if there were any freezing effects on EEA, we assume any freezing effects on EEAs
would be equal across all treatments.  We have added a statement in the Methods section (L. 167-169).

DeForest, J.L. 2009. The influence of time, storage temperature, and substrate age on potential soil enzyme activity in acidic forest soils using MUB-linked substrates and l-DOPA. Soil Biol. & Biochem. 41:1180-1186.

Lee, Y.B., Lorenz, N., Dick, L.K., Dick, R.P. 2007. Cold storage and pretreatment incubation effects on soil microbial properties Soil Sci. Soc. Am. J. 71:1299-1305.

Peoples, M.S., Koide, R.T. 2012. Cosiderations in the storage of soil samples for enzyme activity analysis. Appl. Soil Ecol. 62:98-102.

L165 – Why are only two readings taken and why after only 6 hours? Does the CO2 efflux plateau within 6h?

Response: The 6 h incubation of the plates is directly from the MicroResp™ manual.  But this time, we believed, is based in several papers, one of which is one by Anderson and Domsch (1985).  This paper shows that $CO_2$ is stable in response to glucose for 6 h, but then substrate exhaustion or other factors begin to cause erratic respiration rates at 8-13 h.

Anderson, T.-H., Domsch, K.H. 1985. Maintenance carbon requirements of actively-metabolizing microbial populations under *in situ* conditions.

L241 – "season had no. . .." the first half of this sentence is a bit clumsy and difficult to understand. Perhaps re-word it.

Response: We have reworded this sentence (L. 279-280).

L273-285 – I don't see how the correlation between EEA and CLPP contributes to the overall thesis of the study. These results do corroborate each other and evidence the reliability of the CLPP and EEA data, but, in my opinion do not warrant such a long paragraph in the Results, especially since there is no follow-up discussion points in the Discussion section. I would advise simplifying or leaving this out.

Response: We agree here, and have now reduced this paragraph (L. 323-336) and deleted the duplicated paragraph.

L276 – remove "quite"

Response: This was removed.

L278 – The Fig. S4 does not relate to Nag amine. I think the order of supplementary figures is incorrect and does not correspond to the manuscript. Please check this throughout.

Response: We have corrected the order of the supplemental figures.

L296-308 – all this is a repetition of L273-285. Remove either section and simplify as suggested above.

Response: We removed the duplicate paragraph.

L309 – I do not see any discussion around cover crops and how they affect soil biochemical responses relative to non-cover crop treatments. Increases in soil biochemical functioning may not be a result of plant species diversity per se - rather, cover crops alter soil physical characteristics (e.g. soil moisture
through covering soil in between cash crops) which drive changes in biochemical processes. I would
suggest clarifying the definition of cover crops in the methods and expanding on their relative effects on
soil physico-chemical characteristics in the discussion.

Response: We have added a paragraph discussing the importance of cover crops (L. 378-392).

L321 – I do not see any direct reference to the second hypothesis here. It will make it easier for the
reader to follow if this is done (as you have done in L393).

Response: We now directly refer to the second hypothesis (L. 451-453).

L336 – I would advise some discussion (also in relation to the cover crops) about legumes and soil N.
Increases in microbial biomass may not be driving increased N, rather key-stone microbial species
(rhizobia) may be responsible.

Response: We added some discussion on keystone species, such as legumes (L. 387-389).

L342 – what are the units for "0.1"?

Response: We now have put units after "0.1", "$m^3 m^{-3}$" (L. 482).

L372 – Again, I would suggest making the link to the original hypothesis more explicit (partly done in
L396 but would suggest doing this earlier as well).

Response: We now directly refer to the second hypothesis (L. 451-453).

L416-419 – This is a confusing sentence, please re-word.

Response: We have rephrased this sentence (L. 416-419).

L427-431 – I do not understand this logic. Do you mean to say that using CLPP as a measure of catabolic
evenness in bacterial-dominated soils may not adequately reflect the true microbial catabolic diversity
because (1) bacteria are generalists and use all substrates evenly, and (2) fungi tend to be excluded
through disturbing the soil? If so please re-structure this or explain what you are attempting to say.

Response: We have removed this portion from the discussion because we felt it was not adding much
and to accommodate adding the suggestions from the reviewers.

L431-433 – How does this support the previous statements? 16S rRNA diversity would not necessarily
correspond to catabolic evenness, so cannot be used to firmly support your findings.

Response: We have rephrased these sentences as to indicate this is more speculative (L. 421-423)

Table 1. – Include full stop. To which variables do the units apply to? (e.g. does mg.kg- 1 apply to C:N
ratio?) - make clearer please. Give full descriptions of crop rotation abbreviations in the title.

Response: We have referred to the crop rotation abbreviations in Table 1, and then refer to Table 1 in
the subsequent tables.

---

## Author Comment (AC3) · 8 Sep 2016

**Soil-2016-39 Review 3 comments**

Authors' responses are in blue.

**General comments**:
The article has investigated the effects of temporal plant diversification by rotating crop on soil microbial functions and activities, while many other studies focus on the above-ground spatial biodiversity on soil microbial functions. This study has comprehensively examined a suite of indexes of soil microbial activities (e.g., MBC, MBN, PMC, and PMN) and functions (e.g., EEA and CLPP) over one growing season. The introduction is clear, the methods are easy to follow although they need clarification, and the data interpretation is generally logic. But a couple of issues need to be addressed as follows:

Response: Thank you for this thorough review and kind comments. We really appreciate the time and effort this reviewer put in on their feedback. We have incorporated nearly all their suggestions and we feel it has greatly improved the manuscript.

**Majors**:

1) The main goal of this study was to investigate whether crop rotation can enhance soil microbial biomass and functions. Since the study has examine microbial biomass and functions (e.g., PMC, PMN, and EEA) over one growing season (i.e., in spring, summer, and fall), how confidently we can attribute enhanced microbial activities and functions to crop rotation rather than seasonality or their interactions? Can we confidently say that crop rotation is the main reason for changes in soil microbial functions and activities?

Response: This is an interesting point. We agree here with this reviewer, that in order to have a better understanding (and more confidence) of the season versus crop rotation effects we would need more than one year of data. However, due to the large amount of data collected (or "comprehensive" as this reviewer put it) we were limited to three sampling points. We strategically sampled during times when we would expect there would be differences in these soil microbial responses to look at how season might influence the crop rotation effect on soil microbial biomass and functioning. We are the first, to our knowledge, to have published the catabolic response (or community-level physiological profile, CLPP) on more than one date.

2) Were the crop rotation effects on microbial activities and functions general or unique, considering that all the measurements were for soils that have experienced an extreme drought the year before sampling? What the results will be if the sampling took place in a normal year? Particularly, the authors have discussed that the drying-rewetting effects on soil microbial community in the discussion (L339-359).

Response: We show that this sampling period is very distinct in ways other than high microbial biomass, which is often found at the peak of the growing season (Wardle 2003; Hargreaves & Hofmockel 2013). Since we only sampled once in the summer, we do not know what a "normal" year would look like. Due to the good amount of evidence at hand (L. 485-492, L. 500-504), we

46  attributed the large differences in the summer to this drying-wetting event.  Although, without
47  sampling more times during the summer we do not know for sure.

48  Hargreaves, S. K., and Hofmockel, K. S.: Physiological shifts in the microbial
49      community drive changes in enzyme activity in a perennial agroecosystem,
50      Biogeochem., 117, 67–79, 2013.

51  Wardle, D. A., Yeates, G. W., Williamson, W., and Bonner, K. I.: The response of a three
52      trophic level soil food web to the identity and diversity of plant species and
53      functional groups, Oikos, 102, 45–56, 2003.

**Minors**:

1)  Problems with the reference order in the main text: L27-28, L41-42, L50-51, L120, L157, 380, 381,385-386, 398-399, 401-402, 423

    Response: We preferred to cite papers in the manuscript text chronologically.  SOIL leaves the in-text citation order up to the authors.  "In terms of in-text citations, the order can be based on relevance, as well as chronological or alphabetical listing, depending on the author's preference."  See http://www.soil-journal.net/for_authors/manuscript_preparation.html

2)  L54-58, it is better to introduce the rationale of CLPP method and substrates used in this method, especially for readers who are not familiar with them. More references are needed about how this method works and how widely it has been used in studies to examine soil microbial catabolic functions.

    Response: We now have added more details about the CLPP method and its context in agroecosystems (L. 68-78)

3)  Soil sampling issue. I am confused whether or not the measured soils were sampled under the same crop (maize) in all the three seasons. According the statement in L99-101, it is not the case. Soil sampling took place in April, July, and November, while corn was planted in June and before June some plots may be planted with other crops. Please clarify.

    Response: We now refer to the soils being collected during the same crop phase or year (L. 9, 84, 340) since they technically were not "under" corn during all seasons.

4)  PMC and PMN measurements. How long did the incubation last? It is 6 months in line 122, but it is 120 days in line 129. How was PMC calculated according to cumulative respiration? In addition, how many times were the inorganic N extraction conducted to assess PMN during the entire period of incubation? And how was PMN calculated and what is the reference for it? I cannot find it (L117- 131).

Response: It was 120 days, or 4 months.  We have changed "6" to "4" (L. 139-148).  We further clarified how the cumulative PMC and PMN were calculated (L. 148-151, 192-193).

5) L198, 375 please use the multiple symbol instead of "X" letter.

Response: We have replaced all letter "X's" with multiplication signs (×).

6) Line 2013 should it be "P" rather than "Ps"?

Response: This was changed.

7) L210-211, please clarify the relation between cumulative $CO_2$ respiration and PMC. Otherwise, the statement in these lines is not true.

Response: We removed reference of cumulative respiration and only refer to it as PMC, and they are the same thing.

8) L228-230, the text "increased crop diversity decreasing the $qCO_2$ by 16, 40, 28% in CSW, CSW1, and CSW2…' does not match the results in Fig. 2. In CSW, $qCO_2$ was not always decreased compared to mC (i.e., the control) according to Fig. 2 in spring, summer, and fall. Moreover, is the difference statistically significant?

Response: These were changed, and the significant post-hoc results are displayed in Fig. 2.

9) L291 "negative" should be "negatively"

Response: This was changed.

10) In the discussion, I would like to read the discussion of "crop diversity and soil microbial functions" before "seasonal dynamics and N limitation", as the former is the focus of this study, and they directly answer the two questions asked in the introduction.

Response: We changed the order of these sections.

11) L448-453 the statement does not match Figure. S4. We cannot tell which data are from the less and which are from the more crop rotations from this Figure. Please clarify.

Response: We added the letter 'd' to 'Fig. S5' to signify that this is the panel we were explaining, and hopefully clarify this statement (L. 442).

12) The statements in L399-400 and L411-412 are not clear to me. Which tables or figures can show the results? And what are the indexes of microbial catabolic evenness?

   Response: Thank you for noticing this. We added 'Table 4' after these statements - from where we are drawing on these data (L 399,412).

13) L471-472 as a main finding of this study, it should be discussed in details but it was not. Please specify or remove it.

   Response: We now explain this idea further, and earlier in the Discussion (L. 350-376).

**Figures and tables**

Tables 1 and 2: it is easy to read when all the contents of a table are in one page.

Response: We now have all the components of tables on one page.

Table 3 caption. I do not understand why adding Fig. 5 at the end? What does it mean? This is no Fig. 5 in the manuscript.

Response: This was supposed to be 'Fig. 4'. It has now been changed.

Fig. 1: as mentioned before, cumulative respiration is different from potential mineralizable C, please clarify the relation between them.

Response: We have changed this to PMC.

Figs. 1 and 2: since seasonality has strong effects on soil microbial activity and functions, we need to know how crop rotation effects in each season. It is better to add the multiple comparison of soil microbial activity in each season in these two figures.

Response: We have now shown overall crop rotation post hoc tests in these figures. Because season and rotation did not interact, we felt it was improper to analyze each date individually. Although, we can understand this reviewer's perspective. Sometimes when one factor is dominating ANOVAs it might be better to analyze treatment effects within each date individually. We wanted to highlight interactions, also the direction of treatment trends is rather consistent among each season.

---

## Author Comment (AC4) · 8 Sep 2016

Dear Dr. Bach and Reviewers,

We greatly appreciate the thorough and insightful comments on our manuscript. The addition of these reviewers' comments has substantially improved the manuscript. We are pleased to submit our responses and a revised version of the manuscript. Thank you for allowing this enlightening discussion and chance to re-submit our manuscript.

Sincerely,

Marshall McDaniel and Stuart Grandy